# Vision-centric Token Compression in Large Language Model

**Ling Xing**[1*], **Alex Jinpeng Wang**[2*], **Rui Yan**[1†], **Xiangbo Shu**[1], **Jinhui Tang**[3]

[1]Nanjing University of Science and Technology    [2]Central South University
[3]Nanjing Forestry University

## Abstract

Real-world applications are stretching context windows to hundreds of thousand of tokens while Large Language Models (LLMs) swell from billions to trillions of parameters. This dual expansion send compute and memory costs skyrocketing, making *token compression* indispensable. We introduce VISION CENTRIC TOKEN COMPRESSION (VIST), a *slow–fast* compression framework that mirrors human reading: the *fast* path renders distant tokens into images, letting a **frozen, lightweight vision encoder** skim the low-salience context; the *slow* path feeds the proximal window into the LLM for fine-grained reasoning. A Probability-informed Visual Enhancement (PVE) objective masks high-frequency tokens during training, steering the Resampler to concentrate on semantically rich regions—just as skilled reader gloss over function words. On eleven in-context learning benchmarks, VIST achieves the same accuracy with $2.3\times$ fewer tokens, cutting FLOPs by 16% and memory by 50%. This method delivers remarkable results, outperforming the strongest text encoder-based compression method CEPE by **7.6%** on average over benchmarks like TriviaQA, NQ, PopQA, NLUI, and CLIN, setting a new standard for token efficiency in LLMs. The project is at https://github.com/CSU-JPG/VIST.

## 1 Introduction

Large language models (LLMs) excel at short snippets, yet many real-world tasks, *e.g.*, long-document understanding [1, 2] and question answering [3, 4]—already require inputs far beyond the thousand-token regimes of early GPT-3 [1]. At the same time, parameter counts have leapt from billions to trillions [5, 6, 7]. In this dual squeeze of *longer context & larger models*, **compression shifts from a convenience to a necessity**: without shrinking the input, even the most powerful LLM cannot afford to reason over the information we want it to see.

Psycholinguistics shows that our eyes dance across text: we *fixate* on rare, content-rich words and *skip* almost one-third of high-frequency function words [8, 9, 10]. This *selective-reading* strategy forms a natural *slow-fast* circuit. A *fast visual pass* skims distant, low-salience context to maintain global context, while a *slow cognitive pass* focuses on nearby sentences that matter. (Figure 1 (a)).

Motivated by this circuit, we present VISION CENTRIC TOKEN COMPRESSION (**VIST**), a *slow-fast* token compression framework that mirrors human skimming. As illustrated in Figure 1 (b), VIST first converts loosely relevant long context into images, which are processed by a frozen vision encoder and a trainable Resampler to produce semantically compact visual tokens. These compressed tokens and the main input tokens are then consumed by the LLM. In this *slow-fast* setup, the *vision encoder*

---

* Equal contribution
† Corresponding author

39th Conference on Neural Information Processing Systems (NeurIPS 2025).

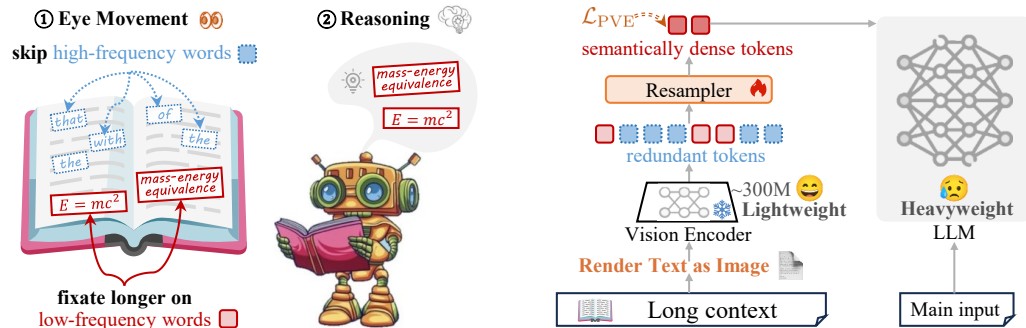

(a) Selective reading strategy based on word frequency in skilled readers.

(b) A visual pathway to unlock LLM context.

Figure 1: Our method VIST adopts a lightweight vision encoder to process loosely relevant long contexts, offering a **more cost-efficient alternative** to full LLM processing. However, the inherent redundancy in long text leads to redundant visual tokens. Motivated by **Selective Reading Strategy** where *low-frequency (content) words receive longer fixations while high-frequency function words are often skipped*, we design Probability-informed Visual Enhancement (*i.e.*, $\mathcal{L}_{PVE}$). This guides the Resampler to prioritize informative content over redundancy, **resulting in a 75% reduction in the number of visual tokens** and yielding semantically dense tokens.

*acts like the human eye*—selectively attending to salient information—while *the LLM functions as the brain*, concentrating on the most informative content for deeper reasoning.

Specifically, the frozen visual encoders (*e.g.*, CLIP [11]) trained on paired image-text data naturally *acquire OCR capabilities* [11, 12], making them a powerful tool for image-based text understanding. However, the inherent redundancy in long text leads to redundant visual tokens. To address the problem, we design **Probability-informed Visual Enhancement** (PVE), a contrastive scheme that enforces Resampler to prioritize informative content over redundancy. Concretely, PVE applies **frequency-based masking strategy** to text token embeddings from the LLM tokenizer, suppressing high-frequency (less informative) text tokens. This semantically rich text supervision guides the Resampler to focus on informative content, bridging the semantic gap between visual and text tokens, and enabling more effective token compression. Unlike previous work [13, 14, 15, 16, 17] that rely on LLMs to compute token-level information entropy for assessing importance, VIST adopts token frequency as a simple yet effective proxy, and further **reveals rare tokens are key contributors to overall semantic meaning** (*cf.* Figure 3 and §5).

VIST leverages a lightweight vision encoder to compress loosely relevant long contexts, offering a cost-efficient alternative to full-scale LLM computation. Furthermore, the vision encoder serves as a *visual text tokenizer*, offering several compelling advantages over traditional text tokenizers. ❶ **Simplified Tokenization.** Text tokenizers rely on complex tokenization rules and vocabulary constraints, typically involving nearly ten human-defined preprocessing steps (*e.g.*, lowercasing, punctuation and stop word removal, and tokenization) [18]. However, the vision encoder processes text more directly by treating rendered text images as visual inputs. ❷ **Vocabulary Bottleneck Mitigation.** Text tokenization, constrained by a finite vocabulary, becomes a bottleneck when scaling to many languages. A larger vocabulary increases memory and computational costs in the embedding matrix and output layer. However, vision encoder eliminates the need for text tokenizers and unifies various languages into a single image format that removes the need for a vocabulary [19, 20]. ❸ **Robustness to Character-Level Noise.** Vision encoders are more resilient to typos and low-level orthographic attacks, as they capture holistic visual patterns rather than relying on discrete token matching [20]. ❹ **Multilingual Efficiency.** While our work focuses on English, visual text tokenizer can reduce the number of tokens compared to traditional text tokenizer for languages (*e.g.*, 62% for Japanese, 78% for Korean, and 27% for Chinese). This reduction is particularly impactful in long-text scenarios. Taken together, *leveraging vision encoders for long-context compression is a promising and worthwhile direction to explore*.

To validate the effectiveness of VIST, we primarily compare with the text-encoder-based token compression counterpart CEPE [21]. VIST requires **2.3× fewer visual tokens** than text tokens for the same input, reducing **FLOPs by 16%** and **memory usage by 50%**. VIST also delivers consistent

gains over CEPE on both In-Context Learning and Open-domain Question Answering tasks, with **average gains of 3.6% across 11 datasets** and **5.7% across 3 datasets**, respectively—highlighting the effectiveness of visual representations for long-context modeling in LLMs.

## 2 Related Work

**Token Compression.** There has been a growing interest in expanding the context window for LLMs. A line of methods leverages LLM itself to compress raw long input. One may classify these works into two principal groups. **i)** *soft prompt-based* methods that adapt LLMs to compress context into fewer tokens [22, 23, 24, 25, 26]. **ii)** *selection-based* methods that remove redundant tokens based on information entropy computed by LLMs [13, 14, 15, 16, 17, 27]. All the long inputs typically need to be handled by the heavy LLMs, which incur high costs. Another line of work [28, 29] augments LLMs with the capacity to memorize previous long context information by external memory bank and retrieve relevant knowledge [30, 31, 32, 33]. Our method is orthogonal to these existing strategies and can be combined with them to achieve longer context length. The most related work is CEPE [21], which employs a lightweight text encoder to handle long contexts and integrates the information into LLM via cross-attention. While CEPE reduces workload on the LLM, it overlooks the redundancy in long text, making it harder for LLMs to effectively allocate attention to key content. In contrast, VIST compresses long text into compact visual tokens guided by high-density semantic text tokens.

**Vision-centric Method.** Text tokenization [34, 35, 36] breaks down text into tokens, serving as a fundamental step in natural language processing. However, tokenization-based methods lack robustness against spelling errors and face vocabulary bottlenecks. A new line of work tackles these issues in a tokenizer-free paradigm [37, 20, 38]. The representative method Pixel [20] renders text as images and learns to reconstruct masked image patches at the pixel level. It demonstrates strong cross-language translation capabilities and tolerance for text perturbation. Along this direction, recent work explores different pre-training objectives [39, 40], *e.g.*, contrastive learning [41], patch-and-text prediction [38]. Despite advancements, these methods overlook long-text scenarios and rely on complicated training pipelines, *e.g.*, OCR-based text understanding [19]. In contrast, VIST directly processes text images by leveraging a vision encoder pretrained on image-text pairs with strong OCR capabilities, and enhancing visual features using enriched text embeddings from LLM tokenizer. An emerging family of multimodal methods [42, 43, 44, 45, 46] leverage visual representations to process text and images together, enabling a wide range of applications involving visually-situated text, *e.g.*, webpage parsing [43], tables images analysis [47], and document understanding [42, 48]. In this work, we explore incorporating long-context information into LLMs from a visual perspective.

## 3 Methodology

In this section, we present our method VIST, which processes long in-context text by a lightweight visual encoder, effectively and efficiently extending the context length of LLMs.

### 3.1 Overall Pipeline

Our VIST, a *slow-fast* compression framework, is designed to efficiently process long texts by mimicking human reading. The *fast visual path* skims distant, low-salience long context via a lightweight vision encoder, while the *slow cognitive path* performs fine-grained reasoning on important content by LLM. As illustrated in Figure 2, the input long text (*i.e.*, $T$ text tokens) is split into two parts: the first $T_e$ text tokens processed in a visual view and the remaining $T_d$ raw text tokens given to LLM, where $T = T_e + T_d$. Specifically, the $T_e$ text tokens are evenly rendered into $M$ images and fed into a frozen vision encoder. Then VIST employs a learnable Perceiver Resampler to compress text-rendered image features into a fixed count of tokens. Such compressed visual tokens are integrated into the LLM via cross-attention for the next-token prediction. The Perceiver Resampler is jointly trained with the LLM during tuning the cross-attentions in an end-to-end manner.

To empower the model with the ability to comprehend dense text in images, we devise **Probability-informed Visual Enhancement** (PVE, §3.4). PVE maximizes agreement between visual features obtained from the Perceiver Resampler and **text token embeddings extracted from LLM tokenizer**. This alignment bridges the global semantic gap between visual tokens and raw text tokens. Furthermore, to address token redundancy, VIST incorporates a **frequency-based masking** mechanism

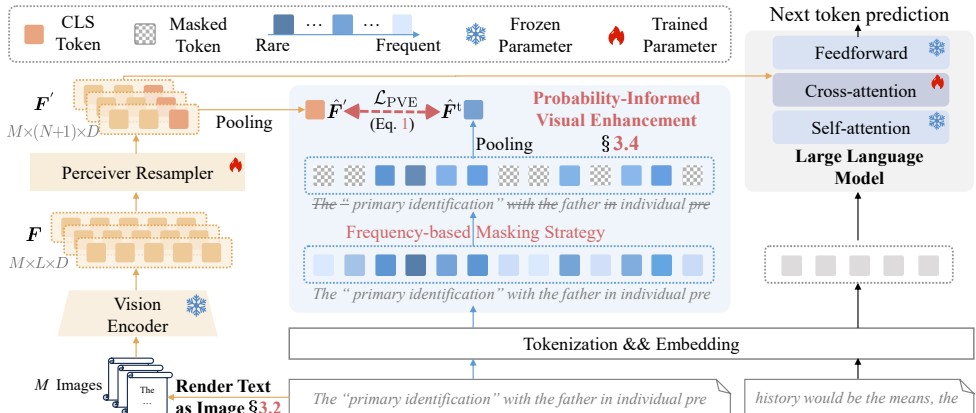

Figure 2: **Overview of VIST**. VIST, a *slow-fast* token compression framework, efficiently processes long texts by mimicking human skimming. First, the *fast visual path* converts long context into images and employs a lightweight vision encoder to capture semantically compact visual features. These features are then integrated into the LLM via cross-attention in the *slow cognitive path*, allowing LLM to focus on salient content for deeper reasoning. To prioritize informative content in text images, VIST employs **Frequency-based Masking** on text token embeddings from text tokenizer, suppressing high-frequency but low-information token ■ (*e.g.*, "the" and "with"). Such refined embeddings guide the Resampler in extracting critical semantics from the images.

within PVE that selectively masks high-frequency, low-information text tokens, thereby improving the information density of the text embeddings. These refined embeddings serve as enriched supervision signals, encouraging visual features to be more compact and semantically meaningful.

## 3.2 Vision-centric Implementation

VIST transforms raw textual data into $M$ uniformly distributed RGB images $\mathcal{X} = \{x_m \in \mathbb{R}^{H \times W \times C}\}_{m=1}^{M}$, where $M$ can be dynamically adjusted based on the length of the input text. Concretely, each image is configured with height $H = 14$, width $W = 3,584$, and $C = 3$ RGB channels, which corresponds to a square color image with a resolution of $224 \times 224$. Text is rendered using a 10px font size and Google *Noto Sans* typeface. If text incompletely fill the image, white empty patches are masked to exclude them from attention score computation and loss calculation. Compared to text tokenizer-based methods, this rendering method does not lead to slower training speeds [44].

## 3.3 Token Reduction

The $M$ text-rendered images are first processed by frozen vision encoder, specifically the ViT-L/14 [11] from OpenCLIP. The extracted features $\boldsymbol{F} \in \mathbb{R}^{M \times L \times D}$ are then fed into a trainable Perceiver Resampler [49], producing a fixed set of $N+1$ visual tokens per image (including a CLS token), denoted as $\boldsymbol{F}' \in \mathbb{R}^{M \times (N+1) \times D}$, where $N = 64$ and $D$ is the feature dimension. During training, raw text data ($T_e = 4096$ text tokens) is rendered onto $M = 28$ images, resulting in $64 \times 28 = 1792$ visual tokens, passed to the cross-attention layer in LLM. This compression reduces the computational complexity of the cross-attention layer within the LLM. Moreover, the number of images $M$ and tokens $N$ can be dynamically adjusted during both training and inference, *allowing* VIST *to flexibly control the compression ratio*. VIST using a lightweight vision encoder, offers a more efficient approach than processing all text tokens directly within the LLM.

## 3.4 Probability-informed Visual Enhancement

In VIST, the frozen vision encoder is pre-trained primarily on general visual data (such as natural images) without exposure to rendered text images. Hence its ability to interpret dense textual information within images is constrained. To alleviate this problem, we develop a novel training objective, named Probability-informed Visual Enhancement (PVE). PVE enhances the understanding capabilities of Perceiver Resampler for rendered text images, enabling them to serve as robust substitutes for traditional text tokenizers.

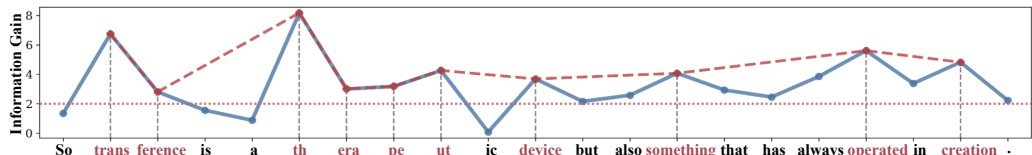

Figure 3: **Token-Level Information Gain** (IG) in sentence "*So transference is a therapeutic device but also something that has always operated in creation.*". The red dashed line masks of 50% the most frequent tokens based on training set statistics. This strategy preserves tokens with higher information gain, while eliminating statistically prevalent but low-value tokens, enhancing semantic density.

**Text-anchored Semantic Consistency.** PVE encourages the Perceiver Resampler to learn a shared embedding space, aligning visual text features $\boldsymbol{F}'$ with text token embeddings from text tokenizer. Concretely, PVE is formulated as a contrastive loss:

$$\mathcal{L}_{\text{PVE}}^{ij} = -\log \frac{\exp(\langle \hat{\boldsymbol{F}}_i', \hat{\boldsymbol{F}}_j^{\text{t}} \rangle / \tau)}{\sum_{k=1}^{B} \exp(\langle \hat{\boldsymbol{F}}_i', \hat{\boldsymbol{F}}_k^{\text{t}} \rangle / \tau)}, \tag{1}$$

where $B$ is batch size and $\hat{\boldsymbol{F}}_i'$ is obtained by applying average pooling to the CLS tokens from $\boldsymbol{F}_i'$. $\hat{\boldsymbol{F}}_j^{\text{t}}$ is the averaged text token embedding after **frequency-based masking** and pooling. $\tau$ is the temperate parameter. Importantly, $\hat{\boldsymbol{F}}_i'$ and $\hat{\boldsymbol{F}}_i^{\text{t}}$ are different representations derived from the same text.

**Frequency-based Masking.** PVE employs text token embeddings as supervision signals to guide the Resampler in extracting textual information from text images. However, long-form text is inherently redundant, where structural components and function words may dominate the token distribution. Such redundancy introduces noise that impedes Resampler from capturing **key semantic content**.

Our solution draws inspiration from Shannon information theory [50], which provides a formal way to quantify the information content of an event or a message. The formula is given by:

$$I(y) = -\log_2 P(y), \tag{2}$$

where $I(y)$ is the information content of event or messages $y$ and $P(y)$ is the probability of $y$. It highlights the inverse relationship between the probability of an event and the information it carries. When applied to tokens in a corpus: **Rare tokens** (low-frequency) are treated as high-information tokens because they often carry domain-specific or contextually important information. **Frequent tokens** (high-frequency) have lower information content because they may serve more structural or grammatical purposes, contributing less to the unique meaning of the text. Figure 3 shows that masking 50% of the most frequent tokens based on corpus-level (*i.e.*, training set) frequency distribution still *preserves most high-information-gain (IG) tokens, ensuring minimal loss of critical information while reducing redundancy*. This aligns with the **selective reading strategies** [10, 51] observed in skilled readers. Based on this principle, we devise frequency-based masking strategy that uses token frequency as a proxy for semantic importance. This strategy masks frequent tokens but low-information tokens to improve the information density of text token embeddings. The importance score for each token is calculated as follows:

$$s_w = \log \frac{|\mathcal{S}|}{1 + \text{count}(w)}, \tag{3}$$

where $|\mathcal{S}|$ denotes the total number of samples, $\text{count}(w)$ is the count of the token $w$ (subword), and $s_w$ is the importance score of token $w$. Token frequency statistics can be easily computed online with negligible overhead or precomputed. Based on the importance score for each token, we apply a 50% masking rate, where *tokens are randomly masked with tokens of lower importance score being more likely to be masked*. This ensures the Resampler prioritizes key content-bearing tokens, learning richer semantic representations and improving its ability to interpret dense text in rendered images.

## 4 Experiment

### 4.1 Experimental Setup

**Pretraining.** We validate VIST with TinyLlama [52]. The frozen vision encoder in our model is ViT-L/14 [11]. To reduce computational overhead, our model employs float16 precision and DeepSpeed Zero-2 with CPU off-loading[53]. Refer to Appendix A for details.

Table 1: Text perplexity on the last 256 tokens of long-context language modeling for ArXiv and Book datasets, PG19, Proof, and Code. $T_e$ is token length for encoder, and $T_d$ is for LLM. † denotes methods with the Resampler and PVE in our VIST. We report the Throughput of each model relative to TinyLlama. $\Delta$ is compression ratio.

| Method | $T_e$ | $T_d$ | ArXiv | Book | PG19 | Proof | Code | Throughput | $\Delta$ | TFLOPs | MEM(GB) |
|---|---|---|---|---|---|---|---|---|---|---|---|
| TinyLlama [52] | - | 4096 | $>10^3$ | $>10^3$ | $>10^3$ | $>10^3$ | $>10^3$ | 1.0× | - | 8.47 | 5.46 |
| Replug [54] | - | 4096 | 3.220 | 15.394 | 14.685 | 3.921 | 3.011 | 0.2× | - | 9.15 | 6.12 |
| Stream [55] | - | 4096 | 3.116 | 15.188 | 14.372 | **2.876** | 2.764 | 1.6× | - | 8.31 | 6.41 |
| ToMe† [56] | 2048 | 2048 | 3.536 | 15.607 | 16.213 | 4.128 | 3.234 | 3.8× | 2.3 | 7.93 | 4.59 |
| FastV† [57] | 2048 | 2048 | 3.491 | 15.711 | 16.016 | 4.216 | 3.111 | 3.8× | 2.3 | 7.88 | 4.59 |
| CEPE* [21] | 2048 | 2048 | 3.071 | 15.619 | **11.737** | 2.888 | **2.151** | 1.8× | - | 8.26 | 4.80 |
| VIST | 2048 | 2048 | **2.993** | **14.973** | 13.205 | 3.057 | 2.247 | 3.8× | 2.3 | **7.72(0.75↓)** | **4.59(0.87↓)** |
| CEPE* [21] | 6144 | 2048 | 3.005 | 14.919 | **11.112** | **2.719** | **2.100** | 2.1× | - | 13.27 | 7.74 |
| VIST | 6144 | 2048 | **2.989** | **14.894** | 12.737 | 3.003 | 2.183 | 5.3× | 2.3 | **11.65(1.62↓)** | **4.94(2.80↓)** |
| CEPE* [21] | 14,336 | 2048 | 3.003 | 14.921 | **10.909** | **2.715** | 2.162 | 3.3× | - | 23.30 | 13.59 |
| VIST | 14,336 | 2048 | **2.965** | **14.815** | 11.933 | 2.971 | **2.032** | 7.6× | 2.3 | **19.52(3.78↓)** | **6.75 (6.84↓)** |

**Competitors. i) Long-context models:** Replug [54] and Stream [55] with TinyLlama [52]. **ii) Text-encoder-based compression method:** To compare the effectiveness of leveraging text tokens *v.s.* visual tokens for processing long contexts in LLM, we implement CEPE* by applying CEPE [21] to TinyLlama [52], replacing the vision encoder in VIST with a lightweight text encoder. All other architectural and training settings are kept identical. **iii) Vision-centric compression methods:** ToMe [56] merges, and FastV [57] prunes visual features from frozen vision encoders. For fairness, we match their compression rates to ours. Directly using these visual features in LLMs leads to high perplexity (>1k) due to the mismatch between visual and text tokens. Thus, we retain the Resampler and PVE in our VIST, denoting ToMe† and FastV†. See Appendix A.5 for more details.

**Pretraining Dataset.** Our pertaining dataset is an official sample of the RedPajama dataset [58], including 1B tokens from seven domains: ArXiv, Book, C4, Commoncrawl, GitHub, StackExchange, and Wikipedia. The training set of the corpus is preprocessed into $4608$ text token sequences, where the first $4096$ text token sequences are fed into the vision encoder (or text encoder for CEPE*) and the remaining $512$ text tokens are provided to LLM.

**Downstream Evaluation.** We primarily evaluate tasks requiring long context processing, revealing vision tokens effectively handle extended context, outperforming previous text-encoder-based model. Comparisons across more methods and datasets are provided in the Appendix A.6.

## 4.2 Long-context Language Modeling

To assess the long-context language modeling (LCM) ability, we evaluate on ArXiv and Book from RedPajama [58] test split, alongside long-context datasets: PG19 [59], Proof [60], and Code [61]. The evaluation metric is perplexity (PPL) over the last 256 tokens of each input. Early tokens (typically farther from the current prediction point) are handled by the vision encoder, while the more recent tokens are passed to the LLM, under the assumption that proximity correlates with relevance.

**Impact of Increased Text Length.** Table 1 summarizes the results across different input lengths. Long-context language modeling can benefit from previous long contextual information. However, TinyLlama [52] supports only fixed-size inputs of 2048 tokens. Beyond this length, its performance drops sharply, with perplexity exceeding $10^3$. In contrast, VIST demonstrates a consistent decrease in perplexity as the input text length increases. Vision-centric token compression models ToMe† and FastV† focus on natural images, where redundancy arises from local visual similarity. However, they are *ill-suited for text redundancy and struggle to preserve key semantic content in text image*, yielding higher perplexity than our VIST. Moreover, VIST obtains the lowest PPL (14.973) on Book datasets, when $T_e$ and $T_d$ are 2048. These results prove that **VIST effectively enhances the capability of modeling long-form language.**

**Comparison on Inference Cost.** In Table 1, VIST renders text into multiple images of size $224 \times 224$. 1024 text tokens need 7 images and VIST requires 56% fewer visual tokens than text tokens for the same input (*i.e.*, compression ratio $\Delta$ is 2.3, from 1024 text tokens to $448 = 7 \times 64$ visual tokens). We report the throughput of each model relative to TinyLlama. VIST achieves comparable performance with text-encoder-based compression model CEPE*, with 16% fewer FLOPs, 50% less memory usage, and higher throughput when processing 16k tokens.

Table 2: In-context learning accuracy averaged across 3 seeds (42, 43 and 44). Green highlights the gain from the additional demos. $n_e$ is the number of demos for encoder and $n_d$ for decoder (LLM).

| Method | $n_e$ | $n_d$ | SST2 | MR | AGN | SST5 | NLUS | NLUI | TREC | TREF | DBP | BANK | CLIN | Avg. |
|--------|------|------|------|------|------|------|------|------|------|------|------|------|------|------|
| TinyLlama [52] | - | 2 | 76.0 | 67.7 | 63.4 | 27.6 | 5.2 | 4.4 | 28.8 | 9.6 | 38.0 | 23.0 | 22.4 | 33.3 |
| TinyLlama [52] | - | 20 | **87.6** | 71.7 | **75.0** | 30.1 | **46.1** | 32.6 | **72.0** | **38.5** | **80.4** | **42.9** | **53.7** | **57.3(24.0↑)** |
| CEPE* [21] | 18 | 2 | 76.9 | **82.3** | 66.9 | 29.1 | 9.6 | 30 | 39.2 | 12.7 | 71.1 | 27.2 | 39.8 | 44.1(10.8↑) |
| VIST | 18 | 2 | 77.7 | 79.2 | 61.5 | **42.7** | 15.6 | **40.6** | 36.5 | 14.6 | 71.9 | 25.0 | 43.8 | 46.3(13.0↑) |
| TinyLlama [52] | - | 50 | **88.6** | 64.8 | 21.4 | 42.5 | **34.2** | 30.4 | **81.1** | **44.7** | 3.4 | **49.7** | 39.7 | 45.5(12.2↑) |
| CEPE* [21] | 48 | 2 | 82.9 | 79.4 | 63.9 | 42.3 | 28.1 | 31.1 | 32.6 | 14.7 | 71.5 | 29.0 | 39.1 | 46.8(13.5↑) |
| VIST | 48 | 2 | 78.9 | **85.2** | **71.9** | **44.4** | 27.2 | **43.1** | 38.3 | 18.4 | **73.1** | 25.4 | **48.1** | **50.4(17.1↑)** |

## 4.3 In-context Learning

We evaluate VIST on In-Context Learning (ICL) tasks across 11 widely-used text-classification datasets: SST2 [62], MR [63], AGN [64], SST5 [62], TREC, TREF [65], DBP [64], NLUS, NLUI [66], BANK [67], and CLIN [68]. Following [21], we randomly sample 250 text examples per dataset. The ICL results in Table 2 are reported as the average accuracy over three random seeds. For VIST and CEPE*, we provide two demonstrations directly to the decoder, while the rest are processed by the encoder. More details in Appendix B.2.

**Results.** Table 2 examines the influence of increasing the number of demonstrations, where $n_e$ is the number of demos for encoder and $n_d$ for LLM. VIST shows a 13% accuracy improvement (from 33.3% to 46.3%) as more demonstrations ($n_e$ is 18) are provided to the visual encoder, showcasing the capacity of LLM to comprehend text within visual signals when integrated with VIST. Furthermore, VIST outperforms CEPE* in average accuracy across all 11 datasets, which indicates **visual-based text understanding can effectively match or even surpass text encoder performance**. Though VIST and CEPE* ($n_e = 18, n_d = 2$) underperform TinyLlama ($n_d = 20$), they *achieve lower computational cost by processing most demonstrations (18) with lightweight encoder*. The performance gap on NLUS, TREC and TREF may stem from high category diversity, where the weak relevance between queries and demonstrations makes the lightweight encoding less effective than using the full LLM for all demos. Notably, the performance of TinyLlama declines with 50 demonstrations due to context window limit, while VIST remains efficient and stable.

## 4.4 Open-domain Question Answering

Open-domain Question Answering (QA) is a challenging task that requires model to generate accurate answers based on retrieved relevant information. Experiments are conducted on three open-domain QA datasets, including TriviaQA [69], NQ [70], and PopQA [71]. We use Contriever [72] to retrieve relevant $k$ passages from Wikipedia, as in CEPE [21]. We prioritize passing the most relevant passages to the decoder to enhance performance. In Table 3, we report the exact match (EM) scores.

**Results.** TinyLlama is limited by a maximum context window of 2048 tokens, restricting it to processing no more than 10 passages at a time. Beyond this limit, performance drops sharply, with EM score falling below 1. For passages $k_d = 10$, the input already approaches or even exceeds the 2048-token limit of TinyLlama, so we truncate the input to avoid performance degradation. When processing 5 extra passages (*i.e.*, $k_e = 5, k_d = 10$), VIST results in an EM score improvement of **3.75** compared to TinyLlama on TriviaQA [69] dataset. Moreover, it even surpasses text encoder-based approach CEPE* under the same input conditions, *e.g.*,

Table 3: **Open-domain QA results.** $k_e$ represents the number of passages provided to the encoder, while $k_d$ denotes the number of passages given to the LLM. We report the exact match score.

| Method | $k_e$ | $k_d$ | TriviaQA | NQ | PopQA |
|--------|------|------|----------|------|-------|
| TinyLlama [52] | - | 10 | 21.45 | 8.45 | 10.79 |
| TinyLlama [52] | - | 15 | < 1 | < 1 | < 1 |
| VIST | 10 | 0 | 21.27 | 8.51 | 10.67 |
| CEPE* [21] | 5 | 10 | 16.41 | 6.09 | 4.92 |
| VIST | 5 | 10 | **25.20(8.79↑)** | **8.71(2.62↑)** | **11.44(6.52↑)** |
| CEPE* [21] | 20 | 10 | 16.56 | 6.75 | 5.78 |
| VIST | 20 | 10 | **25.67(9.11↑)** | **8.81(2.06↑)** | **11.84(6.06↑)** |

delivering an EM score **9.11** points higher on the TriviaQA dataset when $k_e = 20, k_d = 10$. *This enhancement may be attributed to PVE in VIST which leverages enriched text embeddings to guide the Resampler in capturing key semantics.* By emphasizing critical details and filtering out noise from lengthy inputs, our VIST prioritizes relevant information—a crucial factor for success in open-domain QA tasks. In contrast, CEPE* degrades when more passages are provided to the encoder,

Table 4: The effect of visual token count for each image.

| Tokens | ICL | | Open-domain QA | | |
|---|---|---|---|---|---|
| Per Image | TREC | MR | TriviaQA | NQ | PopQA |
| 32 | 32.7 | 78.8 | 19.38 | 7.85 | 8.16 |
| 64 | **36.5** | 79.2 | **25.20** | **8.71** | **11.44** |
| 96 | 32.0 | 79.7 | 14.57 | 7.19 | 4.88 |
| 128 | 32.9 | **87.0** | 20.01 | 7.77 | 8.51 |

Table 6: Ablation on the length of text inputs (in tokens).

| Encoder | ICL | | Open-domain QA | | |
|---|---|---|---|---|---|
| Input Length | SST5 | MR | TriviaQA | NQ | PopQA |
| 1024 | 39.6 | 85.9 | 19.77 | 6.47 | 6.63 |
| 2048 | 39.3 | 73.8 | 22.85 | 8.08 | 9.77 |
| 4096 | **42.7** | 79.2 | 25.20 | 8.71 | 11.44 |
| 6144 | 37.8 | **90.5** | **27.52** | **9.24** | **13.49** |

Table 5: The effect of different masking strategies in PVE. FM is frequency-based masking strategy, RM is random masking. The masking ratio is set to 50%.

| RM | FM | ICL | | Open-domain QA | | |
|---|---|---|---|---|---|---|
| | | NLUS | NLUI | TriviaQA | NQ | PopQA |
| | | 9.9 | 26.4 | 17.14 | 6.51 | 5.72 |
| ✓ | | 8.3 | 30.2 | 24.88 | 8.35 | 10.19 |
| | ✓ | **15.6** | **40.6** | **25.20** | **8.71** | **11.44** |

Table 7: VIST effectively reduces PPL on LCM and improves accuracy on ICL by processing additional context. ‡ means our implementation on Mistral 7B [73].

| Method | LCM | | ICL | |
|---|---|---|---|---|
| | Arxiv | Book | SST2 | DBP |
| Mistral | 2.93 | 12.82 | 89.1 | 93.6 |
| CEPE‡ | 2.83(0.10↓) | 12.64(0.18↓) | 90.8(1.7↑) | 94.2(0.6↑) |
| VIST‡ | **2.82(0.11↓)** | **12.61(0.21↓)** | 92.8(3.7↑) | **95.3(1.7↑)** |

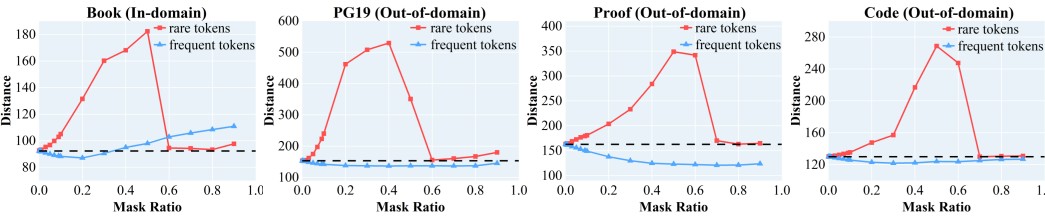

Figure 4: **Effect of frequent *vs.* rare tokens on semantic integrity.** Masking rare tokens (red) significantly disrupts semantic representation, increasing text-visual embedding distance, while masking frequent tokens (blue) has minimal impact, demonstrating that rare tokens are critical for preserving semantic meaning.

as additional passages introduce more noise and redundancy, making it harder to extract relevant answers. We also investigate the effectiveness of using only the fast path of VIST (*i.e.*, vision encoder) on the Open-domain QA task, which requires the model to generate accurate answers based on given relevant passages. Specifically, we feed only the top-10 relevant passages to the visual encoder (*i.e.*, $k_e = 10$), without providing any passages to the LLM directly (*i.e.*, setting $k_d = 0$). Surprisingly, this configuration yields performance on par with TinyLLaMA, despite the latter processing all 10 passages with a much heavier LLM. This demonstrates that the fast path of our method can distill and preserve the critical information from long contexts, providing a compact yet effective representation.

## 4.5 Ablation Study

We explore the effects of ❶ the masking strategy employed in PVE, ❷ the length of text provided to the encoder during training, ❸ the number of compressed tokens in each image (*i.e.*, $N$ in §3.3), and ❹ extension to other LLM. In open-domain QA tasks, the model is fed 10 passages for LLM and 5 for encoder. ICL tasks use a fixed setup of 18 demonstrations for encoder and 2 for LLM. (More details in Appendix F.)

**Number of Tokens in Each Image.** The Resampler transforms image features into a fixed number of visual tokens. Table 4 analyzes the impact of visual token count for each image. Increasing the number of visual tokens reduces compression ratio, but as shown, *a lower compression ratio does not always yield better results*. With 64 visual tokens, the model performs best on 4 out of 5 datasets, whereas 128 tokens only perform best on MR dataset. This discrepancy could be attributed to the trade-off between the amount of information preserved and the noise introduced during compression. Fewer tokens risk losing critical details, while more tokens may retain irrelevant information, which can hinder generalization. These findings emphasize that achieving a balance between compression and information retention is crucial for optimal performance across different datasets.

**Masking Strategy in PVE (§3.4).** Long texts often contain significant redundancy. To address this, we integrate a Frequency-based Masking (FM) strategy into PVE, improving the information density of text token embeddings. Table 5 compares the performance of VIST with FM, w/o FM, and with random masking. Excluding FM causes a notable decline in ICL and open-domain QA performance. This highlights the critical role of information-dense text token embeddings in guiding the visual encoder to capture more semantically meaningful and discriminative features.

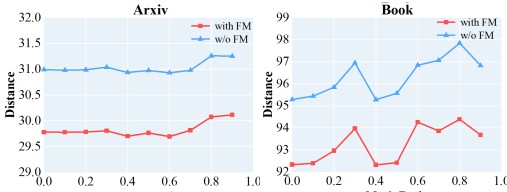

Figure 5: **Impact of Frequency-based Masking on Text-Visual Semantic Distance.** Compared to model without FM, VIST with FM consistently reduces the semantic distance between text token embeddings and visual features, across all masking ratios.

Figure 6: **Visualization of Masking Frequent Tokens.** Though we mask the top 50% most ~~frequent tokens~~ (based on training statistics), **key nouns** are completely preserved, proving that frequent tokens contribute less to semantic meaning.

**In-context Text Length.** Table 6 investigates the impact of in-context text length (measured in text tokens) provided to the visual encoder during training. Our model, trained with a longer encoder input length, generally yields higher EM scores on open-domain QA tasks. This may be because exposure to more lengthy texts during training helps our model better extract key information from extensive contexts, which is crucial for open-domain QA. Table 6 shows that longer training text inputs often boost ICL task accuracy. For instance, the best result on SST5 (42.7) was achieved with an encoder input length of 4096 text tokens, while the highest accuracy on MR (90.5) was obtained with the longest input length of 6144 tokens. Interestingly, we observed that training with an input length of 1024 tokens performed comparably to 2048 tokens, possibly because the total demo length for the encoder was close to 1024 tokens.

**Extension to Other LLM.** Mixture-of-expert models effectively scale model capacity while saving resources. To prove the generality of VIST, we also apply VIST to Mistral 7B [73]. In LCM task, VIST‡ and CEPE‡ process 4096 tokens, compared to the 2048 tokens processed by Mistral. For ICL, VIST‡ and CEPE‡ use 20 demonstrations, while Mistral uses only 2. As shown in Table 7, VIST‡ demonstrates superior performance over CEPE‡, by effectively leveraging additional context. To further validate the scalability of our method, results on larger-scale LLMs are included in Appendix F.

## 5 Discussion

**Exploring Token Frequency as a Proxy for Semantic Importance.** To assess the impact of rare versus frequent text tokens on global semantics across in-domain (Book [58]) and out-of-domain datasets (PG19 [59], Proof [60], and Code [61]), we first calculate importance score for each token using Eq. 3 based on training-set token frequency statistics. Two masking operations are then applied to the text token embeddings: ❶ Masking tokens with high importance scores (red line in Figure 4). ❷ Masking tokens with low importance scores (blue line in Figure 4). The distance between masked text embeddings and visual features is computed.

At low masking ratios (0.0 to 0.4), masking rare tokens causes a sharp increase in distance, while masking frequent tokens has minimal impact and even reduces distance. This suggests that rare tokens carry more critical semantic information, and their removal disrupts the alignment between text and visual tokens. In contrast, frequent tokens may contain more redundant or less informative content, so masking them has little impact or even improves the alignment by reducing noise. These findings support the hypothesis that *token frequency is a reasonable indicator of semantic importance, with rare tokens playing a more pivotal role in preserving semantic integrity*.

**The Effect of Frequency-based Masking on Text-Visual Semantic Gap.** VIST employs Frequency-based Masking (FM) within the Probability-informed Visual Enhancement to improve the semantic richness of text token embeddings. Figure 5 presents the impact of FM on the semantic alignment between text token embeddings and visual features extracted by the Perceiver Resampler. We experimented with random masking ratios (0.0 to 0.9) on text token embeddings and calculated the sum of cosine distances across all test samples in the Arxiv and Book datasets [58]. Across all ratios, VIST *with FM consistently exhibits smaller semantic distances than* VIST *without FM, highlighting FM effectively enhances semantic coherence*.

**Token Redundancy Visualization.** We study the semantic contribution of frequent tokens by selectively masking the top 50% most frequent tokens (based on training set statistics). Figure 6

illustrates key nouns are fully preserved. The masked tokens mainly include function words (*e.g.*, "the", "of"), primarily serving grammatical roles rather than conveying core semantic meaning. This proves FM strategy can **preserve critical semantic information while filtering out less relevant noise, thereby enhancing the ability of the model to focus on meaningful content**.

## 6 Conclusion

In-context learning with longer input sequences remains a prominent yet challenging topic in large language models (LLMs). In this work, we introduce a fully novel perspective to address this challenge by leveraging much lightweight visual encoder. To support longer input sequences in LLMs, we present VIST, a vision-centric token expansion method built upon a visual encoder framework. Our analysis further reveals there exists significant redundancy in text tokens, further validating the effectiveness and efficiency of our vision-encoder-based approach. With these advancements, VIST surpasses text-encoder-based token compression counterparts in both performance and efficiency. In future work, we plan to evaluate VIST across a broader range of downstream tasks and conduct a deeper investigation into text token redundancy.

**Acknowledgement.** This work is supported by the National Natural Science Foundation of China (Grant No. U25A20442, 62332010, 6250074347, 62472208, 62222207, 62427808).

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

# A  Experimental Details

## A.1  Pretraining Details

We provide the data and optimization hyperparameters during the pre-training of VIST in Table 8. We conduct the experiments on NVIDIA V100 GPUs. The cross-attention layer is placed between the self-attention and feed-forward layers in each decoder (LLM) layer. When we first insert the cross-attention layers into the decoder, we initialize the weights of the key, value, and query projection matrices with the respective weights from the self-attention layer of the decoder in the same transformer block. Since the hidden dimension of the encoder $D = 1024$ is smaller than the hidden dimension of the decoder $D'$, $D < D'$, we only copy the first $D$ rows of the key and value projection matrices from the self-attention module to the cross-attention module. The output projection matrix is initialized with norms. The $\tau$ for PVE is 0.07. The weight of PVE is 1.

The rendered text images are shown in Figure 7. We also have several special design choices: (1) we do not attend to the white patches after the end-of-sequence text (following PIXEL [20]); (2) we normalize the input pixel values in each image.

We follow the Perceiver Resampler design used in Flamingo [49], as mentioned in Sec. 3.3. The Perceiver Resampler maps visual features from the frozen Vision Encoder to a fixed number of output tokens. Concretely, this transformer has a predefined number (*e.g.*, 64) of learnable latent vectors as queries, and the keys and values are a concatenation of the visual features with the learnable latent vectors. This mechanism allows the latent input queries to cross-attend to the visual features, producing compact visual representations. Note that the number of output tokens is equal to the number of latent vectors.

## A.2  FLOPs Estimation

We estimate the FLOPs by calculating the number of operations involved in the forward pass for a single input instance. S pecifically, for each instance, we feed $T_e$ tokens into the visual encoder and $T_d$ tokens into the LLM. Only the forward pass is considered; decoding is excluded from FLOPs measurement. This provides a theoretical measure of computational complexity for a static input.

## A.3  Throughput Measurement

Throughput is a practical measure reflecting the actual generation speed on specific hardware, defined as the number of output tokens generated per second during autoregressive decoding. For each input, our VIST processes $T_e$ tokens via the visual encoder and $T_d$ tokens by LLM, and then generates 256 tokens. We calculate Throughput as: $\frac{256}{\text{end-to-end latency}}$, where end-to-end latency refers to the total wall-clock time from receiving the input to the completion of generation (including text tokenization, image rendering, encoding, and decoding). For fair comparison, we ensure all models generate the same number of tokens from inputs of identical length. Note that we normalize the Throughput against TinyLLaMA under the same input-output setting to highlight relative speedups in Table 1.

Figure 7: **Rendered text image.** We demonstrate the result of rendering 256 text tokens onto two 224x224 images.

## A.4 Text Encoder-based Model

In this work, we additionally implement CEPE$^*$. The text encoder in CEPE$^*$ follows the configuration of RoBERTa-large [76], as in CEPE [21]. The encoder contains approximately 400M parameters, while our visual encoder has around 300M parameters. Following CEPE [21], the encoder input tokens are segmented into 256-token chunks for parallel processing and then concatenated together for cross-attention in the LLM.

## A.5 Vision-centric Compression Models

For fairness, we set the number of tokens to merge per layer to 8 for ToMe [56] and the reduction ratio to 75% for FastV [57] to match our compression rate. ToMe merges visual features from the frozen vision encoder, while FastV prunes them.

## A.6 Comparison with Vision-centric and Long-context Models on More Datasets

Owing to space constraints, we present extended comparisons with vision-centric and long-context models on more datasets in the Appendix. We compare our method with vision-centric models (ToMe [56] and FastV [57]) and long-context models (Replug [54] and Stream [55]) on In-context Learning (SST5 and DBP) and open-domain QA (TriviaQA, NQ, and PopQA) tasks, as shown in Table 9. $n_e$ is the number of demonstrations for encoder and $n_d$ for decoder. $k_e$ is the number of passages for encoder and $k_d$ for decoder. To ensure a fair comparison, we provide a total of 20 demonstrations for the in-context learning tasks and 15 passages for the open-domain QA tasks.

Table 8: **Hyperparameters in VIST pretraining.**

| Parameter | Value |
|---|---|
| **Data** | |
| Image size (height, width) | $(224, 224)$ |
| Image mode | RGB |
| Font | Google Noto Sans |
| Font size | 10 |
| **Optimization** | |
| Peak learning rate | $3.0e^{-4}$ |
| Warmup ratio | 4% |
| Learning rate scheduler | Cosine decay [74] |
| Optimizer | AdamW [75] |
| $\beta_1$ | 0.9 |
| $\beta_2$ | 0.999 |
| $\epsilon$ | $10^{-8}$ |
| Mixed precision training | `fp16` |
| Number of steps | 2000 steps |

Table 9: **Comparison with vision-centric and Long-context models.** on In-context Learning (SST5 and DBP) and Open-domain QA (TriviaQA, NQ, and PopQA) tasks.

| Method | $n_e$ | $n_d$ | SST5 | DBP | $k_e$ | $k_d$ | TriviaQA | NQ | PopQA |
|---|---|---|---|---|---|---|---|---|---|
| Replug | - | 20 | 29.1 | 7.3 | - | 15 | 16.07 | 6.32 | 4.77 |
| Stream | - | 20 | 29.7 | 70.5 | - | 15 | 15.13 | 7.11 | 5.12 |
| ToMe[†] | 18 | 2 | 22.0 | 68.0 | 5 | 10 | 14.37 | 6.92 | 4.13 |
| FastV[†] | 18 | 2 | 24.8 | 65.4 | 5 | 10 | 14.69 | 6.08 | 4.96 |
| VIST | 18 | 2 | **42.7** | **71.9** | 5 | 10 | **25.20** | **8.71** | **11.44** |

Since RePlug and Stream do not incorporate an encoder module, all inputs are fed into the decoder. Table 9 shows our VIST achieves the best performance.

### A.7 Extension to other LLM

When applying VIST or CEPE* to Mistral 7B, we leverage DeepSpeed Zero-2 with CPU offloading and use the SDPA attention mechanism. We further optimize the model by quantizing it to 4-bit precision.

## B Downstream Task Evaluation Details

### B.1 Long-context Language Modeling

Following CEPE [21], we evaluate VIST on three long context datasets: PG19 [59], Proof [60], and Code [61], sampling 5000 sequences for each dataset. During inference, VIST optimizes the efficiency and performance of long-text processing tasks by converting text input for vision encoders into images and masking out attention to the empty white patches in such images after the end of the text sequence.

### B.2 In-Context Learning

For our in-context learning experiments, we follow the prompts used in [77, 21] for all datasets. We sample the in-context learning demonstrations from the training set such that each label has an equal number of demonstrations (except for possible remainders).

For CEPE*, the demonstrations for the text encoder are processed in parallel. In contrast, in Table 2, our approach renders all demonstrations into a suitable number of images based on text length, inserts blank lines at the end of each demonstration (similar to the role of "\n" in text processing), and then masks empty, text-free patches to eliminate distractions and enhance focus on meaningful content. More rendering strategies are discussed in Sec. D.

### B.3 Open-domain Question Answering

For CEPE*, additional passages are encoded separately by the text encoder. For our approach, we first consolidate all passages and then render them into an appropriate number of images based on text length. The blank regions at the end of the images are excluded from attention computation.

### B.4 Semantic Distance

In Figure 5 and 4, we calculate the global semantic distance between text token embeddings and visual features from Perceiver Resampler by computing the cosine distance for each sample and then summing the results. For the arXiv and Book datasets, we use the test set for this calculation, while for PG19, ProofPile, and CodeParrot, we randomly sample 100 samples from the test set to compute the distance.

In Figure 5, we fixed the random seed to ensure that the masked content is consistent for methods with and without FM. As the mask ratio increases from 0.1 to 0.5, the distance first rises and then falls.

Table 10: **The impact of masking empty tokens on In-context Learning** (ICL). ICL accuracy averaged across 3 seeds (42, 43 and 44). VIST* does not mask attention to the empty white patches after the end of the text sequence. VIST* and VIST use 2 demonstrations in the decoder, and the remaining demonstrations to the encoder.

| Method | SST2 | MR | AGN | SST5 | NLUS | NLUI | TREC | TREF | DBP | BANK | CLIN | Avg. |
|---|---|---|---|---|---|---|---|---|---|---|---|---|
| **Total Demonstrations** $= 20$ | | | | | | | | | | | | |
| VIST* | 88.7 | 88.9 | 65.2 | 36.9 | 8.4 | 24.8 | 38.8 | 11.4 | 61.2 | 18.0 | 42.0 | 44.0 |
| VIST | 77.7 | 79.2 | 61.5 | **42.7** | 15.6 | **40.6** | 36.5 | 14.6 | 71.9 | 25.0 | 43.8 | 46.3 |

Table 11: **The impact of masking empty tokens on Open-domain QA.** VIST* does not mask attention to the empty white patches after the end of the text sequence. $k_e$ represents the number of passages provided to the encoder, while $k_d$ denotes the number of passages given to the LLM. We report the exact match score.

| Method | $k_e$ | $k_d$ | TriviaQA | NQ | PopQA |
|---|---|---|---|---|---|
| VIST* | 5 | 10 | 21.82 | 7.53 | 8.71 |
| VIST | 5 | 10 | **25.20(3.38↑)** | **8.71(1.18↑)** | **11.44(2.73 ↑)** |

This is may be because random masking at ratios of 0.1 to 0.3 removes more critical information, while at 0.4 and 0.5, it may mask mostly redundant information.

## C  Masking Empty Tokens

We investigate the impact of masking out attention to the empty white patches after the end of the text sequence during both the training and inference stages. By comparing models that attend to these patches versus those that mask them, we analyze their effect on task performance. In Table 10, we present the results for In-context learning tasks, demonstrating that masking empty white patches leads to improved perplexity scores, as the model avoids unnecessary computations on non-informative regions. In Table 11, we show the results for open-domain QA tasks, where masking these patches enhances answer accuracy. These findings highlight the benefits of masking empty patches across different tasks, emphasizing its role in optimizing performance.

Table 12: **Different Rendering Strategies for 18 demonstrations.**

| Method | Description |
|---|---|
| **2-Image Rendering** | All 18 demonstrations rendered into 2 images (high compression). |
| **3-Image Rendering** | All 18 demonstrations rendered into 3 images (balanced compression). |
| **Dynamic Rendering** | All demonstrations rendered into a suitable number of images based on text length. |
| **Dynamic + Blank Lines** | All demonstrations rendered into a suitable number of images with blank lines between demos. |

## D  The Impact of Different Rendering Strategies

In this section, we investigate the impact of different rendering strategies on the in-context learning task. Specifically, we provide the vision encoder with 18 demonstrations and the LLM with 2 demonstrations, and experiment with the following rendering approaches: ❶ **Rendering all 18 demonstrations into 2 images**: This approach tests the ability of the model to handle highly condensed visual representations. ❷ **Rendering all 18 demonstrations into 3 images**: This strategy balances text density and image count, potentially improving readability by achieving a lower compression rate. ❸ **Rendering all demonstrations into a suitable number of images based on text length**: This dynamic approach ensures that each image contains an optimal amount of text, avoiding overcrowding or excessive fragmentation. ❹ **Rendering all demonstrations into a suitable number of images with blank lines inserted between demonstrations**: This method adds visual separation between demonstrations, mimicking the role of paragraph breaks in text processing.

Table 13: **Performance of Different Rendering Strategies on In-context learning tasks.** $k_e$ represents the number of demonstrations provided to the encoder, while $k_d$ denotes the number of demonstrations given to the LLM. We report the average accuracy across three random seeds.

| Method | $k_e$ | $k_d$ | SST2 | MR |
|---|---|---|---|---|
| **2-Image Rendering** | 18 | 2 | 76.0 | 77.7 |
| **3-Image Rendering** | 18 | 2 | 76.6(0.6↑) | 78.2(0.5↑) |
| **Dynamic Rendering** | 18 | 2 | 77.0(1.0↑) | 78.4(0.7↑) |
| **Dynamic + Blank Lines** | 18 | 2 | **77.7(1.7↑)** | **79.2(1.5↑)** |

Table 14: **Comparing visual token counts to text token counts under different settings.**

| Font Size | Visual Token Count | Tokens per Image | Text Token Count | Compression Ratio |
|---|---|---|---|---|
| 10 | 256 | 64 | 218 | 3.41 |
| 9 | 256 | 64 | 240 | 3.75 |
| 8 | 256 | 64 | 270 | 4.22 |
| 7 | 256 | 64 | 310 | 4.84 |

The results, as shown in Table 13, reveal several key insights. The 3-Image Rendering strategy outperforms 2-Image Rendering primarily because it achieves a lower compression ratio, allowing for more detailed visual representations of the text. By distributing the 18 demos across 3 images instead of 2, the text is less densely packed, reducing the risk of information loss and improving the ability of the model to extract meaningful features. The Dynamic Rendering approach, which adapts the number of images based on text length, further improved performance. This indicates that dynamically adjusting the rendering process to match the content complexity is beneficial for maintaining contextual integrity. Finally, the Dynamic+Blank Lines method, which adds blank lines between demos, achieved the highest accuracy. These blank lines introduce clear visual separation, mimicking natural paragraph breaks and acting as visual cues to help the vision encoder distinguish between different text segments. This is particularly important as different demonstrations may belong to distinct categories or contexts. By visually separating them, the model can more easily process each segment independently, reducing the risk of confusion or misinterpretation and improving overall performance. For in-context learning, we adopt Dynamic+Blank Lines as the default rendering strategy.

**Comparing Visual Token Counts to Text Token Counts under Different Settings**. At font size 10, the number of visual tokens (256) from the frozen visual encoder slightly exceeds that of text tokens (218). However, with the Perceiver Resampler, our method reduces the final number of visual tokens per image to just 64. Despite this compression, VIST achieves comparable or even superior performance to the text-encoder-based baseline CEPE, as shown in the Table 14. These showcase that leveraging visual tokens for long-context compression is a promising and worthwhile direction to explore. At font sizes $\leq 8$, the visual token count (256) is even lower than the text token count (310), demonstrating the intrinsic efficiency of the rendered text image. Even at font size 7, where compression is more aggressive, VIST outperforms the CEPE and performs on par with the font size 10 setting. These highlight the robustness of our method to font size variations and further support the effectiveness of using visual tokens for long-context compression.

# E    Comparison on Inference Cost

In Figure 8, we compared the memory usage and computational cost (FLOPs) of three methods—VIST, CEPE*, and TinyLlama—processing inputs of varying lengths. For VIST and CEPE*, when the total input length exceeds 2048 tokens, the excess portion is entirely handled by the encoder. For inputs shorter than or equal to 2048 tokens, the input is split equally between the encoder and the LLM. VIST demonstrates the lowest memory consumption across all input lengths, making it the most memory-efficient method. When processing 32k-length texts, CEPE*and TinyLlama encounter

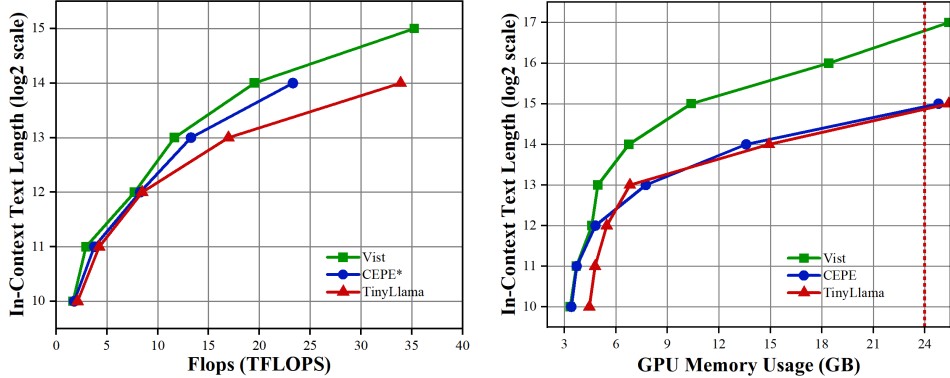

Figure 8: **VIST significantly increases the in-context text length from 16k to 64k at inference stage on a single 24GB RTX 4090 GPU, compared to CEPE\*.** For CEPE\* and VIST the input sequence is partitioned: the last 2048 tokens are processed by TinyLlama, while the remaining tokens are handled by a lightweight encoder.

Table 15: **The effect of different masking strategies in Probability-Informed Visual Enhancement on In-context learning**. We report the accuracy averaged across 3 seeds (42, 43 and 44). All methods use 2 demonstrations in the decoder and 18 demonstrations in the encoder. FM denotes Frequency-based Masking strategy, while RM is random masking strategy. The masking ratio is 50%.

| FM | RM | SST2 | MR | AGN | SST5 | NLUS | NLUI | TREC | TREF | DBP | BANK | CLIN | Avg. |
|----|----|------|------|------|------|------|------|------|------|------|------|------|------|
|    |    | 70.1 | 77.3 | 55.3 | 34.5 | 9.9 | 26.4 | 36.3 | 11.2 | 61.3 | 15.2 | 42.8 | 40.0 |
|    | ✓ | 72.8 | 76.2 | 60.1 | 40.5 | 8.3 | 30.2 | 32.6 | 13.3 | 66.0 | 20.1 | 38.9 | 41.7 |
| ✓  |    | **77.7** | **79.2** | **61.5** | **42.7** | **15.6** | **40.6** | **36.5** | **14.6** | 71.9 | 25.0 | **43.8** | **46.3** |

Out of Memory (OOM) on a 24GB GPU. Compared to CEPE\*, VIST **extends the in-context text length from 16k to 64k tokens** during inference. VIST has a memory usage of 18.4 GB when processing a sequence of 64K, and only encounters an OOM error when processing a 128K sequence.

## F   More Ablation Study Results

In this section, we provide the complete results of our ablation study. Table 15 evaluates the impact of different masking strategies in Probability-informed Visual Enhancement on in-context learning performance. We compare two masking approaches: Frequency-based Masking (FM) and Random Masking (RM), with a fixed masking ratio of 50%. Our method with FM achieves a higher average accuracy compared to without FM or with RM. RM indiscriminately masks text tokens without considering their semantic relevance. This randomness can lead to the masking of critical tokens that are essential for understanding the context, resulting in a loss of important information and a degradation in model performance. While RM may occasionally perform well in certain datasets, its lack of focus on semantically rich tokens makes it less reliable overall. FM specifically masks high-frequency tokens, which are often low in semantic importance, allowing the visual features to focus more effectively on meaningful text tokens. By removing these less informative tokens, FM ensures that **the attention of the model is directed toward text segments that carry more significant contextual information**, thereby improving the quality of the learned representations.

**The Impact of Different Visual Token Counts per Image.** Table 16 investigates the impact of different visual token counts per image on in-context learning performance. The 64-token configuration achieves the highest average accuracy. This suggests that 64 tokens strike an optimal balance between capturing sufficient contextual information and avoiding excessive computational complexity. The 128-token configuration also performs well, particularly on SST2 (89.6) and MR (87.0), indicating that higher token counts can be beneficial for tasks requiring detailed context.

Table 16: **The effect of different visual token count for each image on In-context learning**. We report the accuracy averaged across 3 seeds (42, 43 and 44). All methods use 2 demonstrations in the decoder and 18 demonstrations in the encoder.

| Tokens Per Image | SST2 | MR | AGN | SST5 | NLUS | NLUI | TREC | TREF | DBP | BANK | CLIN | Avg. |
|---|---|---|---|---|---|---|---|---|---|---|---|---|
| 32 | 85.7 | 78.8 | **63.5** | 38.0 | 11.3 | 27.1 | 32.7 | 12.0 | **73.4** | 22.1 | 40.6 | 44.1 |
| 64 | 77.7 | 79.2 | 61.5 | **42.7** | **15.6** | **40.6** | **36.5** | **14.6** | 71.9 | **25.0** | 43.8 | 46.3 |
| 96 | 83.9 | 79.7 | 58.6 | 37.0 | 9.6 | 29.7 | 32.0 | 11.5 | 69.1 | 20.6 | 37.1 | 42.6 |
| 128 | **89.6** | **87.0** | 60.3 | 39.6 | 11.2 | 29.4 | 32.9 | 13.3 | 70.4 | 25.1 | 39.8 | 45.3 |

Table 17: **The effect of the length of text inputs (in tokens) provided to the visual encoder during training**. We report the accuracy averaged across 3 seeds (42, 43 and 44). All methods use 2 demonstrations in the decoder and 18 demonstrations in the encoder.

| Encodr Input Length | SST2 | MR | AGN | SST5 | NLUS | NLUI | TREC | TREF | DBP | BANK | CLIN | Avg. |
|---|---|---|---|---|---|---|---|---|---|---|---|---|
| 1024 | 77.6 | 85.9 | **63.4** | 39.6 | 12.2 | 32.9 | 35.3 | 13.9 | 66.1 | **27.1** | 42.4 | 45.1 |
| 2048 | 78.8 | 73.8 | 61.6 | 39.3 | 12.9 | 34.2 | 35.8 | 13.7 | 67.4 | 26.3 | **44.8** | 44.4 |
| 4096 | 77.7 | 79.2 | 61.5 | **42.7** | **15.6** | **40.6** | **36.5** | **14.6** | **71.9** | 25.0 | 43.8 | **46.3** |
| 6144 | **81.7** | **90.5** | 63.2 | 37.8 | 11.3 | 32.6 | 34.4 | 13.2 | 68.8 | 24.7 | 42.2 | 45.5 |

Interestingly, its average accuracy (45.3) is slightly lower than the 64-token setup, likely due to increased noise or redundancy in the visual representations.

**The Effect of the Length of Text Tokens Provided to the Visual Encoder.** In Table 17, we examine the effect of the length of text inputs (in tokens) provided to the visual encoder during training on in-context learning performance. The results indicate that the 4096-token input length achieves the highest average accuracy (46.3), outperforming other configurations across most datasets, which provides an optimal balance between capturing sufficient context and maintaining computational efficiency. The 6144-token configuration also performs well, particularly on MR (90.5), but its average accuracy (45.5) is slightly lower than the 4096-token setup, likely due to increased noise or redundancy in longer inputs. Similarly, the 1024-token and 2048-token configurations show lower performance, with average accuracies of 45.1 and 44.4, respectively, indicating that shorter input lengths may struggle to provide enough contextual information for the model to perform effectively.

**The Impact of Different Frequency Masking Ratios.** Table 18 examines the impact of different frequency masking ratios (30%, 50%, and 70%) on model performance. We observe that a moderate masking ratio (50%) leads to superior results, indicating the need for a trade-off between redundancy suppression and salient information preservation. Lower ratios (*e.g.*, 30%) may retain redundant or noisy frequency components, potentially distracting the model. In contrast, higher ratios (*e.g.*, 70%) risk discarding salient cues, leading to information loss. The 50% setting effectively suppresses irrelevant signals while preserving essential content, thus enhancing performance.

**Swap Tokens Between the Visual Encoder and LLM.** The relevance scores between passages and the question are available in open-domain QA. We conducted a study to examine how different passage allocation strategies affect performance. Specifically, in row 2 of the Table 19, we assign the $top - k_\mathrm{d} = 10$ most relevant passages to the LLM and the less relevant $k_\mathrm{e} = 5$ to the vision encoder. In row 3 of the Table 19, we swap the assignments (*i.e.*, less relevant passages go to the LLM while more relevant ones are compressed by the vision encoder). The performance remains comparable across both settings (*e.g.*, 8.71 vs. 8.56 on NQ), suggesting that even when highly relevant inputs are compressed, our method preserves essential information, highlighting the effectiveness and robustness of our slow-fast design.

**The Impact of Different Vision Encoders.** We conduct experiments using SigLIP-L [78] as the vision encoder. As shown in the Table 20, SigLIP-L achieves comparable performance to CLIP (ViT-L) within our VIST framework. This demonstrates the robustness and generality of our method across different vision encoders.

**Extension to More LLMs with Different Scales.** We apply VIST on LLaMa2-7B and LLaMa2-13B. Table 21 shows that our VIST with LLaMa2-7B/13B deliver consistent gains over CEPE with

Table 18: **The impact of masking ratio in Frequency-based Masking.**

| Masking Ratio | TriviaQA | NQ | PopQA |
|---|---|---|---|
| 30% | 23.45 | 8.45 | 10.79 |
| 50% | 25.20 | 8.71 | 11.44 |
| 70% | 24.66 | 8.33 | 10.91 |

Table 19: **The impact of swapping tokens between the visual encoder and LLM.**

| Method | $k_e$ | $k_d$ | TriviaQA | NQ | PopQA |
|---|---|---|---|---|---|
| VIST | 5 | $top-10$ | 25.20 | 8.71 | 11.44 |
| VIST | $top-10$ | 5 | 24.68 | 8.56 | 11.35 |

LLaMa2-7B/13B (text-encoder-based baseline) and LLaMa2-7B/13B, highlighting the generalization ability of the proposed method. For the long-context modeling (LCM) task, the encoder is given extended 4096 tokens and the decoder 2048 tokens, with perplexity evaluated over the final 2048 tokens on Arxiv and Book datasets. For the in-context learning (ICL) task, we test on SST2 and DBP, providing 2 demonstrations to the decoder and 18 extra demonstrations to the encoder, and report accuracy.

Figure 9: **Visualization of Masking Frequent Tokens with a 10% masking ratio.**

## G  Visualization of Masking Frequent Tokens

To demonstrate the effectiveness of our frequency-based masking strategy, we conduct a comparative analysis of token masking at different rates (10% in Figure 9 and 50% in Figure 10) based on the frequency statistics derived from the training corpus. The visualization results reveal distinct patterns in the masked token distribution. The most frequently occurring tokens are often common stopwords such as "the", "of", "and", "with", which typically carry less semantic weight. By masking these high-frequency tokens, the model is able to focus on more informative content, thereby improving the overall representation of key semantic features.

## H  Limitations and Future Work

One limitation of our approach lies in the static nature of the masking ratio. A dynamic masking ratio, adjusted based on token distribution, could improve performance. Due to resource constraints, we provide the LLM with a fixed input length of 512 tokens during training. Exploring longer inputs potentially enhances its ability to process more context and capture richer semantics, particularly in scenarios involving more complex or longer texts.

**Multi-turn Dialog with VIST.** To mitigate the inefficiency of re-encoding the entire context in streaming or multi-turn scenarios, a feasible solution is to adopt a lightweight memory caching strategy, inspired by summary accumulation techniques [79, 25], where summary vectors from all segments are concatenated to produce the summary of the entire content. Concretely, at the first dialogue turn, we encode the rendered text images using a frozen lightweight vision encoder followed by a Perceiver Resampler, which effectively compresses them into a small, fixed number of visual tokens (*e.g.*, 64 per image). These compressed visual tokens from Perceiver Resampler act

Table 20: **The impact of different vision encoders.**

| Method | Vision Encoder | Arxiv | Book | SST5 | NLUI |
|--------|----------------|-------|-------|------|------|
| VIST | CLIP | 2.82 | **12.61** | 42.7 | **40.6** |
| VIST | SigLIP | **2.79** | 12.71 | **43.8** | 40.1 |

Table 21: Results of extending our method to LLaMa models of different scales (7B and 13B). Our VIST consistently outperforms CEPE across model sizes, demonstrating strong scalability and generalization.

| Method | LCM | | ICL | |
|--------|-----|------|------|-----|
| | Arxiv | Book | SST2 | DBP |
| LLaMa2-7B | 3.73 | 13.31 | 89.1 | 93.6 |
| CEPE with LLaMa2-7B | 3.11 | 12.78 | 90.2 | 94.0 |
| VIST with LLaMa2-7B | **2.82** | **12.61** | **92.8** | **95.3** |
| LLaMa2-13B | 3.30 | 11.42 | 92.0 | 94.8 |
| CEPE with LLaMa2-13B | 3.01 | 11.03 | 93.1 | 95.4 |
| VIST with LLaMa2-13B | **2.57** | **10.87** | **94.3** | **96.2** |

as summary vectors, capturing the essential semantics of each turn. Instead of discarding them, we cache these memory slots for future use. For each new dialogue turn, we concatenate the new input with all previous cached memory slots and jointly encode them using the same lightweight visual encoder and Perceiver Resampler. This allows the current input to interact with prior context without reprocessing the full raw history and incrementally accumulates dialogue history as a sequence of fixed-size memory slots. Because each turn contributes only a small number of tokens through compact visual representations, the total memory and compute overhead increases slowly—even across many turns. Meanwhile, the memory caching strategy enables each new input to be processed in the context of the entire conversation history. We believe this mechanism makes VIST readily extendable to streaming or interactive tasks, and plan to further explore its application to multi-turn multi-modal dialogue in future work.

Future work could investigate how different font sizes, types, or vision encoders impact performance, particularly in long-context scenarios. While our method currently uses RGB images, future work could explore techniques such as highlighting or bolding important tokens, or even transitioning from RGB to grayscale or binary images, to further reduce computational overhead and improve efficiency.

In this work, we demonstrate that employing a frequency-based masking strategy enhances the information density of text token embeddings. Though directly masking input tokens (*i.e.*, identifying and preserving more critical text tokens during the text-to-image rendering process) could further reduce computational overhead, it may risk degrading performance. Thus VIST employs masking strategy only to the text token embeddings used as supervision signals, while ensuring the integrity of the input data. Certainly, this requires further experimental validation, and it is a promising avenue for exploring a balanced method that optimizes efficiency without compromising performance.

## I  Broader Impacts

In practical applications, the demand for processing long texts is increasingly critical across various domains [2, 80], such as document summarization, legal analysis, medical record processing, and conversational AI. Efficiently handling long texts enables models to capture broader context, leading to more accurate and coherent outputs. In this work, we take a step to address this by leveraging visual tokens, which not only improve the efficiency and performance of current models but also open new avenues for further research in long-text processing. We further reveal high-frequency text tokens often contribute less to semantic meaning. This insight may inspire new algorithms to identify, filter, or simplify low-semantic-contribution tokens, reducing computational complexity, saving resources, and

The latter encompasses the emergence of object and subject, and the **constitution** of **nuclei** of meaning involving **categories**: **semantic** and **categorial fields**. Designating the **genotext** in a text requires pointing out the **transfers** of **drive energy** that can be detected in **ph**one**matic devices** (such as the accumulation and repetition of phonemes or rhyme) and **melod**ic **devices** (such as in**tonation** or **rhythm**), in the way semantic and categorial fields are set out in syntactic and logical features, or in the economy of mimesis (fantasy, the deferment of denotation, narrative, etc.).

Figure 10: **Visualization of Masking Frequent Tokens with a 50% masking ratio.**

enhancing efficiency, especially for large-scale text data. Similar approaches can be extended to other domains, such as image or audio processing [81], to identify and eliminate redundant information, thereby enhancing overall efficiency. This perspective may also complements recent advances in few-shot and zero-shot generalization [82, 83], where models benefit from efficiently focusing on semantically rich cues rather than redundant patterns. Moreover, advances in gaze estimation [84] and visual saliency modeling further provide computational tools to simulate our frequency-mask based mechanism, bridging human perceptual studies and machine learning implementations.

Our work on long-context compression for LLMs improves efficiency, but it also carries potential societal risks. By enabling models to process longer contexts more effectively, it could inadvertently facilitate misuse, such as generating more convincing disinformation or harmful content that leverages extended context. Additionally, compressing long contexts may risk leaking sensitive information or amplifying biases if important contextual nuances are lost. While our method focuses on foundational algorithmic improvements, we acknowledge these risks and encourage future work on responsible deployment practices, including access controls and bias mitigation strategies, to minimize potential harms.

