# OpenReview forum: "Vision-centric Token Compression in Large Language Model"
_NeurIPS.cc/2025/Conference — NeurIPS 2025 spotlight_

### Official Review · Reviewer_jSHm · 2025-06-24

**Clarity:** 3
**Significance:** 3
**Originality:** 3
**Rating:** 5
**Confidence:** 3

**Summary:**

This paper introduces a textual token compression approach using a lightweight vision encoder to process the text contents and a visual resampler to obtain a concentrated text input. The experiments show the proposed method can achieve higher compression ratio than previous method and well maintain the baseline performance.

**Questions:**

1. I recommend that the authors evaluate their method using additional LLMs to strengthen the experimental results and demonstrate the generalizability of their approach.

2. In Table 1, while the reduction in TFLOPs is relatively modest, the throughput shows a significant improvement. I would appreciate clarification on how these metrics were measured, and an explanation for this discrepancy. Understanding the underlying reasons for this phenomenon would help me better interpret the results.

**Ethical Concerns:**

["NO or VERY MINOR ethics concerns only"]

**Final Justification:**

All my concerns have been solved.

**Limitations:**

No, the authors did not discuss the negative societal impact of their work in Appendix J.

**Quality:**

3

**Strengths And Weaknesses:**

## Strengths

1. The proposed method is interesting, and it is impressive to see that it achieves strong empirical results.

2. The masking technique is both simple and effective, leading to a notable improvement in performance.

## Weaknesses

1. The paper aims to compress text tokens for LLMs using a visual encoder. Given this, it is unclear why the authors chose to use an LLM (TinyLlama) instead of a multi-modal LLM (MLLM). Clarification on this design choice would strengthen the paper.

2. Experiments are conducted solely on TinyLlama. The results would be more convincing if validated on a wider range of baseline models.

3. The method for splitting tokens between the visual encoder and the LLM is not clearly described. Additionally, it is not clear how performance is affected if the assignment of text tokens between the visual encoder and LLM is swapped. Further analysis or ablation studies on this aspect would be helpful.

---

> ### Author Rebuttal · Authors · 2025-07-30
>
> We're truly grateful you found our method interesting and its strong empirical results impressive! Thank you for your positive feedback. We'll now address your constructive comments thoroughly.
>
> **Q1: Why choose an LLM over an MLLM?**
>
> **A1:** We sincerely appreciate this insightful question!
> **First,** our primary reason for selecting a text-only LLM was its `superior performance and proven robustness on pure-text tasks`. Since our work focuses specifically on compressing long-context textual inputs, maintaining strong text modeling capability is crucial. While MLLMs are powerful, previous research [1,2] has shown **MLLMs can sometimes experience a performance decline on text-only evaluations compared to LLMs**. Thus, we prioritize the use of a text-only LLM.
>
> **Second,** we entirely agree that **leveraging a multimodal LLM is a fascinating and promising avenue**. In future work, we are keen to explore how MLLMs could benefit from vision-centric compression, particularly for very long textual or truly multimodal inputs. This could involve using visual encoders to enhance their handling of extended contexts or to process documents with rich layouts and embedded images, potentially unlocking new capabilities for both unimodal and multimodal long-context understanding. We will add these in the Appdneix I. Thanks.
>
> [1] WINGS: Learning Multimodal LLMs without Text-only Forgetting. NeurIPS 2024
>
> [2] Qwen2.5-VL Technical Report. ArXiv 2025
>
> **Q2: Extension to other LLMs.**
>
> **A2:** Great suggestion! **First,** we had already provided such kind of experiment, i.e., apply our method to Mistral 7B in Table 7, where our VIST clearly performs better than the CEPE (text-encoder based baseline) in long context tasks.
>
> **Second,** we further **evaluate VIST on LLaMA2 7B**. Below results show that our VIST delivers consistent gains over CEPE, `highlighting the effectiveness of visual tokens for long-context modeling in LLMs`. This experiment will be added in Table 7. Thanks.
> |Method|Arxiv ↓|Book ↓|SST2 ↑|DBP ↑
> |:-:|:-:|:-:|:-:|:-:|
> |LLaMA2 7B|3.73|13.31|89.1|93.6
> |CEPE|3.11|12.78|90.2|94.0
> |VIST (Ours)|$\bf2.82$|$\bf12.61$|$\bf92.8$|$\bf95.3$
> >For the long-context learning task, the encoder is given 4096 tokens and the decoder 2048 tokens, with perplexity evaluated over the final 2048 tokens on Arxiv and Book datasets. For the in-context learning task, we test on SST2 and DBP, providing 18 demonstrations to the encoder and 2 to the decoder, and report accuracy.
>
> **Q3: Method for splitting tokens between the visual encoder and the LLM.**
>
> **A3:** Thank you for pointing this out! Our method VIST is a **slow-fast compression framework** that selectively routes a few highly relevant inputs to the LLM for detailed reasoning, while offloading the majority of less informative ones to a lightweight vision encoder for efficient compression. The token splitting strategy is as follows:
> **1. Training stage**: Given a 4608-token sequence, we feed the first 4096 tokens into the vision encoder and the remaining 512 tokens into the LLM. This setup was described in Sec. 4.1 (lines 194–196).
>
> **2. Inference stage**:
> - For the long-context modeling task, early tokens (typically farther from the current prediction point) are handled by the vision encoder, while the more recent tokens are passed to the LLM, under the assumption that proximity correlates with relevance.
> - For the in-context learning task, since demonstration examples lack explicit importance scores, we randomly allocate the majority (e.g., 48 out of 50) to the visual encoder, with only a small subset (e.g., 2) directly provided to the LLM.
> - For the open-domain QA task, we utilize the relevance scores of passages with respect to the question. VIST assigns a few top-relevant passages to the LLM and compresses the remaining many less-relevant ones using the vision encoder.
>
> We will provide a clearer exposition of these points in Sec. 4. Thanks.
>
> **Q4: Swap tokens between the visual encoder and LLM.**
>
> **A4:** Good comment! The relevance scores between passages and the question are available in open-domain QA. We conducted a study to examine how different passage allocation strategies affect performance. Specifically, in **row 2 of the table, we assign the top-10 most relevant passages to the LLM ($k_d=10$) and the less relevant $k_e=5$ to the vision encoder**.
> In row 3, we swap the assignments (i.e., less relevant passages go to the LLM while more relevant ones are compressed by the vision encoder). `The performance remains comparable across both settings` (e.g., 8.71 vs. 8.56 on NQ), suggesting that even when highly relevant inputs are compressed, our method preserves essential information, highlighting the effectiveness and robustness of our slow-fast design. We will append this content to Appendix G. Thanks.
>
> Method|$k_e$|$k_d$|TriviaQA|NQ|PopQA
> :-:|:-:|:-:|:-:|:-:|:-:
> VIST|5|top-10|25.20|8.71|11.44
> VIST|top-10|5|24.68|8.56|11.35
>
> **Q5: Clarification on how FLOPs and Throughput were measured, and an explanation for their discrepancy.**
>
> **A5:** Here's a detailed explanation:
> - **FLOPs Estimation.** We estimate the FLOPs by calculating the number of operations involved in the **forward pass for a single input instance**. Specifically, for each instance, we feed $T_e$ tokens into the visual encoder and $T_d$ tokens into the LLM. **Only the forward pass is considered**, decoding is excluded from FLOPs measurement. This provides a theoretical measure of computational complexity for a static input.
> - **Throughput Measurement.** Throughput is a practical measure reflecting the **actual generation speed** on specific hardware, defined as the number of output tokens generated per second during autoregressive decoding. For each input, our VIST processes $T_e$ tokens via the visual encoder and $T_d$ tokens by LLM, and then generates 256 tokens. We calculate Throughput as: $\frac{256}{\text{end-to-end latency}}$, where *end-to-end latency* refers to the total wall-clock time from **receiving the input to the completion of generation** (including text tokenization, image rendering, encoding, and decoding). For fair comparison, we ensure all models generate the same number of tokens from inputs of identical length. Note that we normalize the Throughput against TinyLLaMA under the same input-output setting to highlight relative speedups in Table 1.
> - **Explanation for the Discrepancy.** The discrepancy arises because FLOPs measure the total theoretical floating-point operations in the forward pass, which primarily reflects arithmetic workload but does not account for crucial factors like memory access patterns or the iterative nature of autoregressive decoding. In contrast, Throughput measures actual generation speed based on wall-clock time encompassing all practical runtime factors. Thus, **Throughput captures practical inference speed more comprehensively than FLOPs alone**.
>
> We will add these in Appendix B. Thanks.
>
> **Q6: Negative societal impact.**
>
> **A6:** Thanks for your careful review. We will add Appendix J with the following content:
> Our work on long-context compression for LLMs improves efficiency, but it also carries potential societal risks. By enabling models to process longer contexts more effectively, it could inadvertently facilitate misuse, such as generating more convincing disinformation or harmful content that leverages extended context. Additionally, compressing long contexts may risk leaking sensitive information or amplifying biases if important contextual nuances are lost. While our method focuses on foundational algorithmic improvements, we acknowledge these risks and encourage future work on responsible deployment practices, including access controls and bias mitigation strategies, to minimize potential harms.
>
> We appreciate again your thoughtful review and we hope we addressed your concerns. Please let us know if you'd like any further information!

---

> > ### Comment · Reviewer_jSHm · 2025-08-01
> > **Reponse to Rebuttal**
> >
> > Thanks for your response, which addresses my concerns. I will update my score to 5.

---

> > > ### Author Response · Authors · 2025-08-01
> > >
> > > Thank you sincerely for your time and dedication throughout this review process! We truly appreciate your thorough evaluation of our manuscript and rebuttal.

---

### Official Review · Reviewer_pQBS · 2025-06-29

**Clarity:** 3
**Significance:** 2
**Originality:** 3
**Rating:** 5
**Confidence:** 3

**Summary:**

This paper introduces VIST (Vision-centric Token Compression), a framework for efficiently extending the context length of large language models (LLMs).

VIST draws inspiration from human selective reading, where readers focus on informative, low-frequency words and skim over high-frequency ones. Following this slow-fast strategy, VIST converts distant, low-salience text into images processed by a lightweight, frozen vision encoder, while keeping proximal context as raw text tokens.

To enhance compression quality, the authors propose Probability-Informed Visual Enhancement (PVE), a training objective that masks high-frequency, low-information textual tokens and aligns visual features with the remaining semantically rich text embeddings. This enables the vision encoder to produce semantically dense visual tokens.

Experiments on long-context language modeling demonstrate that VIST reduces the number of tokens by 2.3× compared to the original text, while also cutting FLOPs by 16% and memory usage by 50% relative to the text-encoder-based compression baseline CEPE. Additionally, VIST consistently outperforms CEPE on most in-context learning and open-domain question answering tasks.

**Questions:**

- Have you considered how to deal with multi-turn dialog using VIST?

**Ethical Concerns:**

["NO or VERY MINOR ethics concerns only"]

**Final Justification:**

The authors provide additional results on image font size and the choice of vision encoder, demonstrating the robustness of the proposed method. They also include a discussion on handling multi-turn dialog scenarios. These additions effectively address my concerns, and thus I decide to raise my rating to 5.

**Limitations:**

yes

**Quality:**

3

**Strengths And Weaknesses:**

### Strength
- The idea of compressing long textual context by rendering them as images is novel and interesting.
- VIST demonstrates strong performance across three key task categories—long-context language modeling, in-context learning, and open-domain question answering—consistently outperforming text-encoder-based compression methods while clearly reducing computational costs.
- Ablation study demonstrates the effectiveness of the proposed PVE training strategy, and the analysis on the visual-text embedding distance provides valuable insights into the frequence-based masking strategy.
### Weaknesses
- The number of visual tokens in VIST depends on several factors, including the resolution of the rendered text images, font size, and the number of tokens mapped to each image. However, the current manuscript does not provide a systematic analysis comparing visual token counts to textual token counts under different settings of these factors. Such an analysis is essential to better understand the advantage of image rendering in compressing long textual context.
- VIST employs an encoder-decoder architecture to handle long-context inputs. While effective in static settings, this design is ill-suited for tasks involving continuous or streaming inputs, such as multi-turn dialogue. In these scenarios, the entire context must be repeatedly re-encoded with each new input, leading to significant inefficiencies.
- The choice of vision encoder is a critical component of the VIST framework, as it directly affects the quality of visual token representations. However, the current manuscript does not explore the impact of different vision encoders.

---

> ### Author Rebuttal · Authors · 2025-07-30
>
> We're thrilled you found our vision-centric compression idea interesting and the strong performance of VIST! Thank you for your comprehensive review. We'll now address your valuable feedback.
>
> **Q1: Comparing visual token counts to text token counts under different settings.**
>
> **A1:** Good suggestion!
> For the **same input text**, increasing image resolution produces more patches from the vision encoder. But most extra patches are blank and can be easily removed (i.e., the number of meaningful visual tokens remains unchanged). We therefore fix the image resolution to 224×224 and focus our analysis on the two factors that directly affect visual token count: **font size** and the **number of tokens mapped to each image**.
>
> To quantify the relationship between visual and text token count, we randomly sampled 1,000 documents from PG-19 [1] and computed the average number of text tokens that correspond to 256 visual tokens (A 224×224 rendered text image yields 256 visual tokens via the frozen vision encoder CLIP ViT-L/14 [2]). The term *Tokens per Image* refers to the number of tokens mapped to each image by Perceiver Resampler. The *Compression Ratio* is defined as: $\text{Compression Ratio} = \frac{\text{Text Token Count}}{\text{Tokens per Image}}$.
>
> ### Font Size
> Font Size|Visual Token Count|Tokens per Image|Text Token Count|Compression Ratio
> :-:|:-:|:-:|:-:|:-:
> 10|256|64|218|3.41
> 9|256|64|240|3.75
> 8|256|64|270|4.22
> 7|256|64|310|4.84
>
> 1. At font size 10, the number of visual tokens (256) from the frozen visual encoder slightly exceeds that of text tokens (218) for the **same input text**. However, with the Perceiver Resampler, our method reduces the final number of visual tokens per image to just **64**. **Despite this compression, VIST achieves comparable or even superior performance to the text-encoder-based baseline CEPE** [3], as shown in the table below. These showcase that leveraging visual tokens for long-context compression is a promising and worthwhile direction to explore.
> 2. **At font sizes ≤ 8, the visual token count from the frozen visual encoder (256) is even lower than the text token count (310)**, demonstrating the intrinsic efficiency of the rendered text image.
> 3. Even at **font size 7**, where compression is more aggressive, **VIST outperforms the CEPE and performs on par with the font size 10 setting**. These highlight the robustness of our method to font size variations and further support the effectiveness of using visual tokens for long-context compression.
>
> Method|Font Size|Tokens per Image|TriviaQA|NQ|PopQA
> :-:|:-:|:-:|:-:|:-:|:-:
> VIST (Ours)|10|64|$\bf25.20$|$\bf8.71$|$\bf11.44$
> VIST (Ours)|7|64|24.68 |8.46|11.13
> CEPE|-|-|16.41|6.09|4.92
>
> ### Number of Tokens per Image
> We had already conducted ablations on the number of tokens mapped to each image in Table 4. Higher Tokens-per-Image values correspond to lower compression rates. Interestingly, **a lower compression ratio (i.e., more tokens per image) does not always improve performance**. In fact, with only 64 visual tokens, VIST achieves the best performance on 4 out of 5 datasets in Table 4, outperforming configurations with 96 or 128 tokens. This suggests that our visual tokens preserve critical semantics even under strong compression and underscores the importance of **balancing redundancy suppression and salient information retention**.
>
> We will add these analyses to Appendix E. Thanks.
>
> **Q2: Multi-turn dialog using VIST.**
>
> **A2:** Thank you for the thoughtful comment! To mitigate the inefficiency of re-encoding the entire context in streaming or multi-turn scenarios, a feasible solution is to adopt a `lightweight memory caching strategy`, inspired by summary accumulation techniques [4,5], where summary vectors from all segments are concatenated to produce the summary of the entire content. Concretely:
> - At the first dialogue turn, we encode the rendered text images using a frozen lightweight vision encoder followed by a Perceiver Resampler, which effectively compresses them into a small, fixed number of visual tokens (e.g., 64 per image).
> - These compressed visual tokens from Perceiver Resampler act as summary vectors, capturing the essential semantics of each turn. Instead of discarding them, we cache these `memory slots` for future use.
> - For each new dialogue turn, we concatenate the new input with `all previous cached memory slots` and jointly encode them using the same lightweight visual encoder and Perceiver Resampler. This **allows the current input to interact with prior context without reprocessing the full raw history**.
>
> This approach **incrementally accumulates dialogue history as a sequence of fixed-size memory slots**. Because each turn contributes only a small number of tokens through compact visual representations, the total memory and compute overhead increases slowly—even across many turns. Meanwhile, the memory caching strategy enables **each new input to be processed in the context of the entire conversation history**.
>
> We believe this **memory caching mechanism** enables VIST to `naturally extend to streaming scenarios without requiring any architectural modifications`. As a promising direction, we plan to explore its application to multi-turn dialogue in future work.
>
> It is worth noting that the challenge of repeated context encoding in streaming settings is not unique to VIST. The text-encoder-based baseline CEPE suffers from the same limitation. However, **CEPE encodes raw text into a significantly larger number of tokens**, which leads to higher memory consumption and poorer scalability in multi-turn or streaming scenarios. To fully benefit from the **memory caching strategy** that alleviates re-encoding overhead, `CEPE would require modifications to more aggressively compress its text token representations`. We will include this discussion in Appendix I. Thanks.
>
> **Q3: The impact of different vision encoders.**
>
> **A3:** We appreciate your valuable suggestion! We conduct experiments using SigLIP-L [6] as the vision encoder. As shown in the table below, SigLIP-L achieves comparable performance to CLIP (ViT-L) within our VIST framework. This demonstrates the robustness and generality of our method across different vision encoders. These results will be added to Appendix G. Thanks.
> |Method|Vision Encoder|Arxiv ↓|Book ↓|SST5 ↑|NLUI ↑
> |:-:|:-:|:-:|:-:|:-:|:-:
> |VIST|CLIP|3.67|$\bf15.26$|42.7|$\bf40.6$
> |VIST|SigLIP|$\bf3.42$|15.31|$\bf43.8$|40.1
>
> >For long-context modeling task, we allocate 2048 tokens to the visual encoder and 2048 to the LLM, and report perplexity over the last 2048 tokens on the ArXiv and Book datasets. For in-context learning, 18 demonstrations are provided to the visual encoder and 2 to the LLM, with accuracy reported on SST5 and NLUI.
>
> We sincerely appreciate your constructive suggestions. We hope we have addressed all of your concerns. Please let us know if you require any additional information.
>
> [1] Compressive transformers for long-range sequence modelling. ICLR 2019
>
> [2] Learning transferable visual models from natural language supervision. ICML 2021
>
> [3] Long-Context Language Modeling with Parallel Context Encoding. ACL 2024
>
> [4] Recurrent memory transformer. NeurIPS 2022
>
> [5] Adapting Language Models to Compress Contexts. EMNLP 2023
>
> [6] Sigmoid Loss for Language Image Pre-Training. ICCV 2023

---

> > ### Comment · Reviewer_pQBS · 2025-08-02
> > **Response to Author Rebuttal**
> >
> > I thank the authors for the detailes rebuttal. My concerns have been basically addressed and I believe the additional results and discussion would strength the paper. Therefore, I am raising my rating to a 5.

---

> > > ### Author Response · Authors · 2025-08-02
> > >
> > > We deeply appreciate your positive response and the raised rating! It's very encouraging to know that the clarifications and results addressed your concerns.

---

### Official Review · Reviewer_wnpj · 2025-07-01

**Clarity:** 2
**Significance:** 2
**Originality:** 4
**Rating:** 4
**Confidence:** 5

**Summary:**

The inference cost of LLMs directly scales with the number of input tokens. This has propelled a lot of work around ways to compress the input tokens or process the input tokens intelligently, while undergoing minimum performance loss. This paper proposes an interesting idea of parsing long context input in LLMs as multiple text images and passing them through a vision encoder which has OCR capabilities. THe vision encoder output can then be projected to the LLM embedding layer, and processed with some text input (finegrained input) to get the overall prediction. While the approach in novel and interesting, the empirical results are not that convincing as I detail in my review.

**Questions:**

See the weakness section.

Overall, while the proposed ideas are interesting, it looks like unfortunately, the empirical observations are not strong enough,

**Ethical Concerns:**

["NO or VERY MINOR ethics concerns only"]

**Final Justification:**

I have updated my score as the concerns have been resolved.

**Limitations:**

yes

**Quality:**

3

**Strengths And Weaknesses:**

Strengths:
- The idea is very interesting, and quite different from the standard attention and other thresholding based token compression works that I have read.
- I really like the motivation of the approach around how humans read: finegrained + coarse attention.
- The end to end approach makes sense, along with various sub-components like frequency based masking.

Weakness:
My key issue with the paper is that the empirical results show it’s limited success in practice, while being an interesting idea.

- First, in Table 1, the authors evaluate perplexity on the last 256 tokens in a long context input. Note that the preceding 2k tokens (T_d) are processed without any compression i.e. via standard LLM forward pass. Now, it is not clear whether the T_e tokens play an important role in determining the perplexity, especially given that the immediately preceding tokens are anyways being processed via standard forward pass. This issue can even be seen in the fact that as T_e i.e. the context is being increased (from 2048 to 14336), the final perplexity doesn't seem to improve that much.
- Moreover, in Table1, the gains over the CEPE text encoder based baseline doesn’t seem to be significant enough, both in terms of flops reduction or performance.
- Table 2: Authors seem to work with small in-context learning tasks, that too for text classification (like SST2). It would be better to try out modern long context  analysis tasks, needle in hay stack tasks, etc. Nonetheless, even in the setting the authors considered for Table 2, the performance drop with token reduction is indeed significant (like from 57 to 46). This really raises questions about efficacy of the approach or the claims.
- Table 3 is a bit hard to digest. If the input context to standard model (TinyLlama) is with 15 passages, why does EM score drop below 1. Shouldn’t your implementation ignore the earlier input if the context length is limited.
- Similarly, in Table 3, on NQ and PopQA, it looks like the additional set of passaages (k_e) do not really bring significant performance improvement beyond that of what you get with just processing recent past input (k_d).

---

> ### Author Rebuttal · Authors · 2025-07-30
>
> Glad to see the idea and motivation resonated with you! Thank you for your insightful review. We provide point-to-point response below.
>
> **Q1: The impact of extended $T_e$ visual tokens on perplexity (PPL).**
>
> **A1:** Good comment! In Table 1, we follow the experimental setup of CEPE [1] and compute PPL over the last 256 tokens.
> 1. Despite a substantially increased context fed to visual encoder, the PPL improvement is less pronounced than anticipated. This likely because the additional context is distant from the prediction targets and offers limited semantic cues. Notably, `this phenomenon is not unique to our current setup`—similar trends have been widely observed in previous studies [2-4] on pure LLMs.
> 2. To more directly evaluate the impact of the $T_e$ tokens processed by visual encoder on PPL, we conducted an additional experiment. Specifically, with a total sequence length of $T_e + T_d$ tokens, the initial $T_e$ tokens are processed by the visual encoder and the last $T_d$ tokens are passed to decoder. We then track the PPL of the final $T_d=2048$ tokens conditioned on the visual tokens. Results on all three datasets show our VIST outperforms CEPE. Critically, **PPL consistently decreased with increasing $T_e$ (e.g., from 13.83 to 12.67), highlighting the positive impact of extended visual context**.
>
> |Method|$T_e$|$T_d$|Arxiv|Book|PG19|
> |:-:|:-:|:-:|:-:|:-:|:-:|
> |CEPE|2048|2048|3.99|15.85|15.04|
> |VIST (Ours)|2048|2048|3.67|15.26|13.83|
> |CEPE|6144|2048|3.61|15.49|14.70|
> |VIST (Ours)|6144|2048|3.22|14.94|13.22|
> |CEPE|14,336|2048|3.38|15.00|14.16|
> |VIST (Ours)|14,336|2048|2.97|14.42|12.67|
>
> **Q2: Gains over CEPE in terms of FLOPs reduction and performance in Table 1.**
>
> **A2:** We would like to clarify that our VIST **achieves a favorable trade-off between compression efficiency and performance, especially in long-context scenarios**. Specifically:
> - **Efficient semantic compression:** In Table 1, VIST achieves powerful semantic compression, converting text into **2.3× fewer visual tokens** than CEPE uses text tokens. `Despite this substantial compression, VIST not only remains competitive with CEPE but often outperforms it`, demonstrating the semantic richness of our compressed visual tokens for long-context modeling.
> - **Stronger performance with the compressed visual tokens:** In experiments evaluating PPL relying on lightweight encoder outputs (Table in Q1), VIST consistently achieves superior performance compared to CEPE by effectively leveraging compressed visual tokens (e.g., 12.67 vs. 14.16). A lower PPL in this evaluation highlights the strength of our VIST learned visual tokens in capturing essential context.
> - **Practical efficiency gains:** In longer contexts, the advantage of VIST becomes more evident (e.g., for $T_e$ = 30,720, VIST achieves an **8.32 TFLOPs** reduction compared to CEPE). However, as FLOPs only quantify the theoretical cost of a single forward pass, we also report **Throughput** to better reflect practical, end-to-end latency in Table 1, where VIST shows a significant improvement of with longer inputs (e.g., a 7.3× increase).
> - **Future potential:** Our current method is an initial exploration of leveraging visual tokens for long text compression. We believe there is still significant room for improvement, and plan to explore stronger compression strategies, better token selection mechanisms, and more efficient training objectives to further boost both performance and efficiency.
>
> In summary, while CEPE serves as a strong baseline, our method offers distinct advantages in compressibility and inference efficiency without sacrificing model accuracy. We hope the reviewer finds this trade-off meaningful for long-context modeling. Thanks.
>
> **Q3: Evaluation on more long-context tasks (e.g., Needle-in-a-Haystack).**
>
> **A3:** Great suggestion! In Table 2, we followed the in-context learning setup from CEPE [1] and PCW [5]. We agree that incorporating more long-context tasks would further strengthen our claim.
>
> We conducted a Needle-in-a-Haystack evaluation to assess the ability of VIST to process long contexts using compressed visual representations. Specifically, we **encode the entire long context using the vision encoder and provide only the question to the LLM**. For TinyLLaMA, the LLM directly processes the entire content. We vary the total context length from 1k to 32k tokens in increments of 4k, and control the depth percent from 0% to 100% in steps of 10%.
>
> Due to the character limit and inability to include figures at this stage, we report **average accuracy** across all settings. VIST achieves 82.60%, significantly outperforming CEPE (67.12%) and TinyLLaMA model (19.53%). These results demonstrate that VIST is more effective at identifying and utilizing sparse, relevant information from long and noisy inputs. We will include the detailed results in Appendix G. Thanks.
>
> Method|Average Accuracy
> :-:|:-:
> TinyLLaMA|19.53%
> CEPE|67.12%
> VIST (Ours)|82.60%
>
> **Q4: In Table 2, the performance drops with token reduction from 57 to 46.**
>
> **A4:** We would like to explain the performance gap and clarify our VIST is effective.
> 1. **Balancing between performance and computation. a)** TinyLLaMA with 20 demonstrations achieves 57.3% by fully processing all demos with the LLM, incurring high computational cost. In contrast, our method only uses 2 decoder inputs and `compresses the other 18 demonstrations into a small set of visual tokens via a lightweight visual encoder`. Although this leads to some performance drop (46.3%), it greatly reduces LLM workload and improves efficiency. **b)** Notably, CEPE processes all 18 additional demonstrations through a lightweight text encoder without any compression, resulting in a higher token count than our method. Despite this increased input, it still `suffers from a more severe performance drop` (57.3% → 44.1%). This comparison proves that our compression approach **VIST is superior at retaining essential information while achieving appealing efficiency gains**.
> 2. **Notable gains from leveraging visual tokens.**
> By processing 18 additional demonstrations via our lightweight visual encoder, our approach **achieves a notable +13% gain** over TinyLLaMA baseline, boosting accuracy from 33.3% to 46.3%. This result clearly demonstrates the effectiveness of leveraging visual tokens for enhancing context and improving performance.
> 1. **Robustness to long context lengths.** TinyLLaMA's performance drops to 45.5% when using 50 demos due to context overflow, while our method remains robust and even improves to 50.4%. This highlights our method’s stability under growing input length.
>
> **Q5: EM score below 1 when TinyLLaMA processes 15 passages.**
>
> **A5:** Sorry for the confusion. In Table 3, for $k_d=10$ passages, the input **already approaches or even exceeds** the 2048-token limit of TinyLLaMA, so we truncate the input to avoid performance degradation. Applying the same truncation to 15 passages yields results **nearly identical** to the 10-passage case, as seen in the table. Thus, we do not truncate at 15 passages to highlight the performance drop (EM score<1) when the context limit is exceeded. We will clarify this point in Sec. 4.4 of the paper. Thanks.
> |Method|$k_d$|TriviaQA|NQ|PopQA|
> |:-:|:-:|:-:|:-:|:-:|
> |TinyLLaMA|10|21.45|8.45|10.79|
> |TinyLLaMA|15|21.46|8.45|10.81|
>
> **Q6: Performance improvement on NQ and PopQA in Table 3.**
>
> **A6: First,** in Table 3, we assign the most relevant passages to the LLM, while routing the less relevant $k_e$ passages to the lightweight vision encoder. On NQ and PopQA, increasing $k_e$ alone leads to limited performance gains, **likely because most passages are only weakly relevant to the question**. When we assign the top-10 most relevant passages to the visual encoder ($k_e=10$) and additionally give the less relevant $k_d=5$ passages to the LLM, the **performance gain remains limited on NQ and PopQA, even though the heavier LLM processes these extra less relevant passages directly.**
>
> Method|$k_e$|$k_d$|NQ|PopQA
> :-:|:-:|:-:|:-:|:-:
> VIST (Ours)|10|0|8.51|10.67
> VIST (Ours)|10|5|8.56|11.25
>
> **Second,** to more clearly demonstrate the capacity of VIST to preserve salient information while discarding irrelevant content, we **conduct an additional analysis where only the top-$k_e$ passages are processed by the vision encoder (with $k_d = 0$)**. As shown in the table, `performance improves significantly as more passages are provided to the visual encoder.` Remarkably, when all top-10 passages ($k_e = 10$) are processed by the vision encoder, **VIST achieves performance comparable to TinyLLaMA**, even though TinyLLaMA processes the same 10 passages using a heavier LLM. These results underscore the effectiveness of VIST’s visual pathway in compressing long-context inputs while preserving critical information. We will include these results in Table 3. Thanks.
>
> Method|$k_e$|$k_d$|TriviaQA|NQ|PopQA
> :-:|:-:|:-:|:-:|:-:|:-:
> VIST (Ours)|5|0|16.18|5.45|5.79
> VIST (Ours)|10|0|21.27|8.51|10.67
> TinyLLaMA|0|10|21.45|8.45|10.79
>
> Finally, we would like to sincerely thank you for your careful review and valuable comments. We hope we addressed all your concerns. Please let us know if you'd like any further information.
>
> [1] Long-Context Language Modeling with Parallel Context Encoding. ACL 2024
>
> [2] Can perplexity reflect large language model’s ability in long text understanding? ICLR 2024
>
> [3] LONGNET: Scaling Transformers to 1,000,000,000 Tokens. ICLR 2023
>
> [4] What is Wrong with Perplexity for Long-context Language Modeling? ICLR 2025
>
> [5] Parallel context windows for large language models. ACL 2023

---

### Official Review · Reviewer_SgyK · 2025-07-01

**Clarity:** 3
**Significance:** 3
**Originality:** 3
**Rating:** 5
**Confidence:** 4

**Summary:**

This work proposes a vision-centric token compression method for large language models (LLMs), named VIST, which renders long textual contexts as images and employs a lightweight vision encoder for compression. In addition, a probability-informed visual enhancement (PVE) strategy is introduced to guide the compression process toward semantically rich regions, thereby improving the alignment between visual and textual features. Experimental results demonstrate the effectiveness of the proposed approach across multiple benchmarks and tasks.

**Questions:**

1.	The authors mention that a lower compression ratio does not always lead to better results. Could this be due to potential side effects introduced by the frequency-based masking strategy?
2.	Does adopting frequency masking with different ratios contribute to improved performance?
3.	Would the proposed approach remain effective if only the fast path were applied directly to processed the whole rendered text as an image
4.	As mentioned in the weaknesses, can the proposed method scale to large-scale LMMs? In addition, is the Perceiver Resampler pre-trained offline or jointly trained with the LLM in an end-to-end manner?

**Ethical Concerns:**

["NO or VERY MINOR ethics concerns only"]

**Final Justification:**

The authors resolved our previous concerns regarding the methodology's clarity and generalization ability during the rebuttal and discussion period. We therefore assign an Accept rating as our final decision.

**Limitations:**

yes

**Paper Formatting Concerns:**

There are no significant formatting issues observed in this paper.

**Quality:**

3

**Strengths And Weaknesses:**

Strengths:

1.	The paper is well-written and effectively presents the motivation and method in a concise and organized manner.
2.	The idea of rendering text as images for long-context compression is both novel and intuitively appealing.
3.	The experimental results are comprehensive and convincingly demonstrate the effectiveness of the proposed vision-centric compression strategy.

Weaknesses:

1.	The scalability and generative capabilities of the proposed approach across different-sized LLMs are not sufficiently validated. We suggest including experiments with LLaMA-2 models at multiple scales, particularly since LLaMA-2 serves as the baseline in CEPE. This would provide a more meaningful comparison and better demonstrate the generalization ability of the proposed method, beyond the current results with only TinyLLaMA-1.1B and Mistral-7B.
2.	The manuscript briefly mentions the Perceiver Sampler but does not provide a detailed explanation of its architectural design. Including a more thorough description of its structure would enhance clarity and help readers better understand its role and details.
3.	The training pipeline of VIST lacks sufficient clarity. Specifically, it remains ambiguous whether the Perceiver Resampler is pre-trained offline using the loss $L_{PVE}$ or jointly optimized with the LLM during tuning the cross-attentions. We suggest explicitly stating the training strategy to avoid confusion and to improve reproducibility.

---

> ### Author Rebuttal · Authors · 2025-07-30
>
> We greatly appreciate your kind words about the novelty of our vision-centric compression idea, the comprehensiveness of our experimental validation, and the clarity of our paper! Thank you for your positive and encouraging feedback. We will carefully address each of your thoughtful comments below.
>
> **Q1: Extension to LLaMA-2 models at multiple scales beyond TinyLLaMA-1.1B and Mistral-7B.**
>
> **A1:** Great suggestion! Given our computational resources, we apply VIST on LLaMa2-7B. Below results show that `our VIST with LLaMa2-7B delivers consistent gains over CEPE with LLaMa2-7B (text-encoder-based baseline) and LLaMa2-7B`, highlighting the generalization ability of the proposed method. This experiment will be added to Table 7. Thanks.
> |Method|Arxiv ↓|Book ↓|SST2 ↑|DBP ↑
> |:-:|:-:|:-:|:-:|:-:|
> |LLaMa2-7B|3.73|13.31|89.1|93.6
> |CEPE|3.11|12.78|90.2|94.0
> |VIST (Ours)|$\bf2.82$|$\bf12.61$|$\bf92.8$|$\bf95.3$
> >For long-context modeling task, the encoder is given extended 4096 tokens and the decoder 2048 tokens, with perplexity evaluated over the final 2048 tokens on Arxiv and Book datasets. For in-context learning task, we test on SST2 and DBP, providing 2 demonstrations to the decoder and 18 extra demonstrations to the encoder, and report accuracy.
>
> **Q2: Perceiver Sampler structure.**
>
> **A2:** Thanks for your careful review! We follow the Perceiver Resampler design used in Flamingo [1], as mentioned in Sec. 3.3. The Perceiver Resampler maps visual features $X_v$ from the frozen Vision Encoder to a fixed number of output tokens. Concretely, this transformer has a predefined number (e.g., 64) of learnable latent vectors as queries, and the keys and values are a concatenation of the visual features $X_v$ with the learnable latent vectors. This mechanism allows the latent input queries to cross-attend to the visual features, producing compact visual representations. Note that the number of output tokens is equal to the number of latent vectors. We will provide a detailed explanation of Perceiver Resampler in Appendix B. Thanks.
>
> [1] Flamingo: a visual language model for few-shot learning. NeurIPS 2022.
>
> **Q3: Clarify whether the training strategy is end-to-end.**
>
> **A3:** Sorry for this confusion. The Perceiver Resampler is `jointly trained` with the LLM during tuning the cross-attentions in an `end-to-end manner`. We will emphasize this in the Sec. 3.1. Thanks.
>
> **Q4: Whether the frequency-based masking (FM) strategy affect the relationship between compression ratio and performance?**
>
> **A4:** We appreciate the thoughtful question!
> 1. To investigate whether the FM strategy introduces side effects that affect the relationship between compression ratio and performance, we conducted an ablation using `our VIST model without FM`. **The observed trend remains consistent—a lower compression ratio does not always yield better performance, even in the absence of FM.**
> 2. One plausible explanation is that **lower** compression ratios preserve more tokens, which helps retain finer semantics. However, **the marginal benefit may plateau or even decline due to increased redundancy and reduced abstraction**. These results underscore a critical trade-off between compactness and semantic richness, independent of the FM strategy. Thanks.
>
> Tokens per image|TriviaQA|NQ|PopQA
> :-:|:-:|:-:|:-:
> 32|15.45|5.45|3.79
> 64|$\bf17.14$|$\bf6.51$|$\bf5.72$
> 96|16.06|5.33|4.91
> 128|16.35|5.62|4.88
> >Higher Tokens-per-Image values correspond to lower compression rates.
>
> **Q5: The impact of frequency masking with different ratios.**
>
> **A5:** Good suggestion! We conducted an ablation study to examine the impact of different frequency masking ratios (30%, 50%, and 70%) on model performance. We observe that a `moderate masking ratio (50%) leads to superior results`, indicating the need for a trade-off between redundancy suppression and salient information preservation. **Lower** ratios (e.g., 30%) may retain redundant or noisy frequency components, potentially distracting the model. In contrast, **higher** ratios (e.g., 70%) risk discarding salient cues, leading to information loss. The 50% setting effectively suppresses irrelevant signals while preserving essential content, thus enhancing performance. The experimental results will be added to Appendix G.
>
> Masking Ratios|TriviaQA|NQ|PopQA
> :-:|:-:|:-:|:-:
> 30%|23.45|8.45|10.79
> 50%|$\bf25.20$|$\bf8.71$|$\bf11.44$
> 70%|24.66|8.33|10.91
> >All models are evaluated on open-domain QA task using exact match score on TriviaQA, NQ, and PopQA datasets. For a fair comparison, we provide 10 passages to the LLM and 5 passages to the visual encoder across all settings.
>
> **Q6: Would the proposed approach remain effective if only the fast path were applied?**
>
> **A6:** Good comment! We investigate the effectiveness of using only the fast path on the Open-domain QA task, which requires model to generate accurate answers based on given relevant passages. Specifically, we **feed only the top-10 relevant passages to the visual encoder** ($k_e=10$), without providing any passages to the LLM directly (i.e., setting $k_d = 0$). Surprisingly, this configuration yields `performance on par with TinyLLaMA`, despite the latter processing all 10 passages with a much heavier LLM. This demonstrates that **the fast path of our method can distill and preserve the critical information from long contexts, providing a compact yet effective representation.** We will add these results to Table 3. Thanks.
>
> |Method|$k_e$|$k_d$|TriviaQA|NQ|PopQA|
> |:-:|:-:|:-:|:-:|:-:|:-:|
> |VIST (Ours)|10|0|21.27|8.51|10.67|
> TinyLLaMA|0|10|21.45|8.45|10.79
>
> We appreciate again your thoughtful review and hope we addressed all your concerns. Please let us know if you'd like any further information.

---

> > ### Comment · Reviewer_SgyK · 2025-08-01
> >
> > Thank you for your detailed review, which addresses most of our concerns and can improve the manuscript's clarity. However, ​​one remaining issue​​ is whether the approach can scale to LLMs larger than 7B. We believe this is important to demonstrate the method's practical capability. Testing on a model like LLaMA2-13B would suffice to show the scaling potential of the proposed approach and may not be substantially more costly than using LLaMA2-7B. ​​Should you address this point, we would raise our score to Accept; otherwise, we will maintain it as Borderline Accept.​​

---

> > > ### Author Response · Authors · 2025-08-06
> > > **Extension to LLaMA2-13B**
> > >
> > > We appreciate your thoughtful suggestion! To address the concern regarding scalability, we extend VIST to LLaMA2-13B. As shown in the results, VIST shows `clear improvements over both CEPE with LLaMA2-13B (text-encoder-based baseline) and LLaMA2-13B`. These findings provide further empirical support for the **scalability and robustness of our approach**.
> > >
> > > |Method|Arxiv ↓|Book ↓|SST2 ↑|DBP ↑
> > > |:-:|:-:|:-:|:-:|:-:|
> > > |LLaMa2-13B|3.30|11.42|92.0|94.8
> > > |CEPE|3.01|11.03|93.1|95.4
> > > |VIST (Ours)|$\bf2.57$|$\bf10.87$|$\bf94.3$|$\bf96.2$
> > > >For long-context modeling task, the encoder is given extended 4096 tokens and the decoder 2048 tokens, with perplexity evaluated over the final 2048 tokens on Arxiv and Book datasets. For in-context learning task, we test on SST2 and DBP, providing 2 demonstrations to the decoder and 18 extra demonstrations to the encoder, and report accuracy.
> > >
> > > ----
> > >
> > > We hope this additional evidence effectively addresses your concern and highlights the scalability of VIST.
> > > We greatly appreciate your continued engagement and valuable feedback!

---

> > > > ### Comment · Reviewer_SgyK · 2025-08-07
> > > >
> > > > Thank you for providing the experiments. All our concerns have been addressed, and we will increase the rating.​​

---

> > > > > ### Author Response · Authors · 2025-08-08
> > > > >
> > > > > We greatly appreciate your thoughtful feedback and the improved rating. We're pleased to hear that our clarifications and results have fully addressed your concerns.

---

### Author Response · Authors · 2025-08-09

**To all reviewers:**

We express our sincere gratitude to all reviewers for their valuable time and thorough assessment of our manuscript. In response, we have carefully addressed each concern raised, and provided point-to-point clarifications which shall be integrated into the new version of our manuscript.

We are gratified by the positive feedback from all reviewers, particularly regarding the novelty and interest of the idea (**Reviewer SgyK, wnpj, pQBS, and jSHm**), the strong and comprehensive empirical results (**Reviewer SgyK, pQBS, and jSHm**), and the clarity of our writing (**Reviewer SgyK**).

Foremost among the concerns is the robustness of our compressed vision tokens. We provide a detailed and thorough analysis to validate their effectiveness and reliability across the following aspects:
- **Effect of extended $T_e$ visual tokens on perplexity (PPL):** We track the PPL conditioned solely on the $T_e$ visual tokens, which consistently decreases as $T_e$ grows, highlighting the positive impact of extended visual context.
- **Performance with only the fast visual path:** With only the fast visual path, our method attains performance on par with TinyLLaMA at substantially lower computational cost, underscoring the efficiency and fidelity of our compressed visual tokens.
- **Token swapping between visual encoder and LLM:** Swapping key tokens to the lightweight visual encoder maintains performance comparable to that of LLM processing, confirming that visual tokens can reliably encode important context.
- **Generalizability across vision encoders:** When integrating our method with alternative vision encoders (SigLIP), we observe consistent effectiveness.
- **Scalability to larger LLMs:** Applying our method to larger LLMs (LLaMA2-7B and LLaMA2-13B) results in clear gains compared to both CEPE (text-encoder-based compression baseline) and the vanilla LLaMA2 model.

Other questions, such as the Perceiver Sampler structure, clarification on end-to-end training, the impact of masking ratios in FM, Needle-in-a-Haystack evaluation, analysis for comparing our method with CEPE/TinyLlama, comparison of visual and text token counts under different settings, efficient handling of streaming inputs, LLM selection rationale, token splitting methods, and FLOPs/throughput calculations, have also been addressed accordingly.

For more details, please refer to our responses to each reviewer. We have strived to address each of your concerns.

Sincerely yours,

Authors

---

### Decision · Program_Chairs · 2025-09-17

**Decision:**

Accept (spotlight)

**Comment:**

This paper proposes vision-centric token compression for long-context LLMs by rendering distant tokens as images processed by a lightweight encoder while keeping proximal text as standard tokens, guided by probability-informed masking. Reviewers praised the novelty, solid gains on sufficient benchmarks, and thorough ablations, but initially questioned scalability beyond TinyLLaMA and clarity of training and splitting strategies. Authors responded with LLaMA2-7B/13B results, needle-in-haystack tests, and clarifications, satisfying all reviewers. Minor remaining issues (streaming dialog, encoder choice) are acknowledged. Based on the above reason, this paper is recommended for acceptance.